

# Scrutinising an inscrutable bark-nesting ant: Exploring cryptic diversity in the *Rhopalomastix javana* (Hymenoptera: Formicidae) complex using DNA barcodes, genome-wide MIG-seq and geometric morphometrics

Wendy Y. Wang[1] and Aiki Yamada[2]

[1] Lee Kong Chian Natural History Museum, National University of Singapore, Singapore, Singapore
[2] Systematic Zoology Laboratory, Department of Biological Sciences, Graduate School of Science, Tokyo Metropolitan University, Tokyo, Japan

Corresponding author
Wendy Y. Wang,
nhmwyw@nus.edu.sg

## ABSTRACT

Overlooking cryptic species diversity has grave implications on assessments of climate change impacts on biodiversity, ecosystems and organismal populations. Discriminating between cryptic species has long been challenging even for seasoned taxonomists, as interspecies morphological differences are often indiscernible by visual observation. Multi-disciplinary methods involving genetic analyses in conjunction with quantitative morphological data, should therefore be used to investigate boundaries between cryptic species. We adopted an integrated approach combining analyses of mitochondrial COI barcodes, a genome-wide dataset obtained *via* multiplexed inter-simple sequence repeats (ISSRs) genotyping by sequencing (MIG-seq), and geometric morphometrics to investigate species divergences in the inscrutable *Rhopalomastix javana* species complex. Objective clustering of COI suggested five putative molecular species units divergent from each other by thresholds within 4.2–10.6% uncorrected pairwise distance. Phylogenetic analyses based on concatenated MIG-seq data also recovered and strongly supported the monophyly of five major lineages in agreement with COI clusters. Co-ancestry analyses based on MIG-seq data using fineRADstructure resolved variable patterns of admixture linked to geography, and potential genetic drift within some putative species. Geometric morphometric analyses of specimen images further detected statistically significant differences in at least one of three anatomical aspects (Head, Meso, Profile) between all pairs of putative species. Head shape (full-face view) was determined to be the most informative character for species diagnosis, with relatively high classification accuracy. Thin-plate spline deformation grids highlighted areas of high variation between species in each shape for deeper taxonomic scrutiny. The presence of species from multiple distinct lineages existing in near-sympatry firmly demonstrates that *R. javana* comprises more than one closely-related species, but exact species boundaries are difficult to ascertain. Differences in elevation and its associated abiotic effects on ant adaptations and reproductive phenology may contribute to restricting gene flow and maintaining species boundaries between

sympatric populations of the *R. javana* complex. We further assess the advantages and limitations of geometric morphometrics as a taxonomic tool. Despite its drawbacks, our combined approach has helped draw important insights on cryptic diversity in *R. javana*, and also identified gaps of knowledge that await address. Results from this study will inform and prime future in-depth taxonomic investigation on the *R. javana* complex, including formal descriptions and establishment of the five putative species.

# INTRODUCTION

The practice of identifying and differentiating species forms the bedrock of myriad research areas in the biological sciences. It bears critical implications on conservation, biodiversity monitoring, pest and invasive species management, even environmental impact assessments and climate change (*Wilson, 2004*). Conventional methods of species delimitation based on morphology are challenging or almost impossible in morphologically cryptic species—where species boundaries are ill-defined by external appearances alone—even for the most seasoned experts. Cryptic species are apparently common and widespread among most animal phyla (*Fišer, Robinson & Malard, 2018*), especially invertebrates including insects. Overlooking cryptic species and relying wholly on morphospecies-based assessments tend to result in severe underestimation of the true scale of biodiversity loss with global climate change (*Bálint et al., 2011*).

Moreover, losing biodiversity entails diminishing genetic diversity—this has dire ecological consequences for populations, communities and ecosystems, since organismal resilience to environmental changes due to climate change hinges on the latter. The recognition of cryptic species is also important in facilitating more accurate interpretations of organismal responses to broad environmental gradients. For example, in a study on maximum thermal tolerance limits of cryptic rolled-leaf beetles species along an elevational gradient, *García-Robledo et al. (2016)* debunked earlier global analyses which determined thermal tolerance as a plastic trait within species. Instead, the study clearly showed that thermal tolerance limits were fixed within discrete species and haplotypes. Those results implied that species initially thought to have broad elevational distributions and variable thermal tolerances, may actually comprise cryptic species each with narrow elevation distributions. The failure to identify cryptic species ultimately undermines the protection of organisms predisposed to greater extinction risk because of their lower tolerance of changing temperatures with global warming.

Multi-disciplinary approaches in assessing species boundaries—often termed "integrative taxonomy"—are frequently adopted when it comes to research on cryptic species. Integrative taxonomy usually involves combining morphological, molecular, ecological and/or behavioural lines of evidence to support the delimitation of species-level lineages (*Padial et al., 2010*). Studies on cryptic species typically involve empirical methods based on both genetic (DNA) markers and quantitative morphological data. Amongst the

latter, geometric morphometrics (GM) has consistently demonstrated its utility as a more powerful alternative to traditional morphometric methods (*Adams, Rohlf & Slice, 2004*; *Zelditch, Swiderski & Sheets, 2012*; *Klingenberg, 2016*; *Katzke et al., 2018*).

Unlike conventional morphometrics that relies on independent linear measurements, GM focuses on analysing shapes or configurations of points as whole entities, after removing the potentially confounding effects of size and rotation. The notion of "shapes" in GM comprises not only of mere outlines, but can also encompass information on the arrangement of specific features relative to one another in a broader context. Hence, GM methods can help detect subtle morphological differences not discernible by the naked eye. When used in conjunction with genetic methods, GM can be an effective tool in testing species boundaries and identifying candidate characters for species diagnosis.

Integrated taxonomic frameworks involving both DNA and GM have been adopted in assessing cryptic species or species complexes in many insects (*e.g.*, beetles—*Huang & Knowles, 2016*; bugs—*Masonick & Weirauch, 2020*). Such studies have slowly gained traction in recent times, though conventional morphometrics still remains the dominant choice over GM, particularly for integration with molecular data (see *Prebus (2021)* and *Schär et al. (2022)*). Most research on species delimitation using combined DNA and GM methods in the order Hymenoptera (*i.e.*, ants, bees and wasps) focus on comparing wing shapes and/or wing vein configurations. Shapes of specific body parts or male genitalia are sometimes also analysed, but to a smaller extent. Moreover, such studies mainly investigate cryptic species in bees (*e.g.*, *Aytekin et al., 2007*; *Bonatti et al., 2014*; *Duennes et al., 2017*) and parasitoid wasps (*e.g.*, *Chroni et al., 2018*; *Rudoy et al., 2022*), but not ants. While some studies have tested the efficacies of GM methods in species discrimination for the ants (*e.g.*, *Seifert, Ritz & Csősz, 2014*), only one study (at the time of writing) has examined in depth the combined DNA-GM approach in ant taxonomy, *i.e.*, *Wagner et al. (2017)*.

More recent research involving GM methods to differentiate ant species centred on genera with relatively large body sizes and visibly distinct traits—these cannot be considered strictly as "morphologically cryptic". *Katzke et al. (2018)* analysed the wings of the fossil genus *Titanomyrma* Archibald et al. (2011), while *Tozetto & Lattke (2020)* described variation in male genitalia of *Dinoponera* Roger, 1861 using GM methods. Both genera are considered 'giant ants' in layman terms. *Samung et al. (2022)* used GM to distinguish between three species of the moderately large-sized trap-jaw ant genus *Odontomachus* Latreille, 1804. The three focal species are morphologically very distinct (*Satria et al., 2015*), and easily separated visually without need of GM analysis. In general, these studies fail to demonstrate the full potential of GM as a taxonomic tool for detecting and quantifying indistinct morphological differences within and between cryptic ant species.

Cryptic species are ubiquitous amongst the ants. Numerous studies have demonstrated the existence of cryptic species from different genera using either genetic data alone (*e.g.*, *Chialvo et al., 2018*; *van Elst et al., 2021*), or—more frequently—a combination of genetic and morphological information (*Schlick-Steiner et al., 2006*; *Csősz et al., 2014*; *Hosoishi & Ogata, 2019*; *Schär et al., 2022*). Cryptic diversity has been examined in economically-important groups of ants, including fire ants of the genus *Solenopsis*—some

of which are infamous global invasives (*Chialvo et al., 2018*)—and the black garden ant *Lasius niger* (Linnaeus, 1758), an invasive pest widespread in Europe (*Schär et al., 2022*). Morphological crypsis has also been observed in endemic ants, such as honeypot ants of the genus *Myrmecocystus*, which are keystone species of arid ecosystems in North America (*van Elst et al., 2021*). The only study to date that specifically synthesized DNA and GM methods, focused on resolving cryptic diversity in the *Tetramorium caespitum* (Linnaeus, 1758) complex (*Wagner et al., 2017*). Comprehensive analyses of both DNA and GM data provided robust evidence supporting 10 European species in the complex. Despite its apparent utility in revealing cryptic diversity, however, the combined DNA-GM approach has not been applied on any other ant species following the 2017 study.

With their diminutive size and nondescript appearance, the inscrutable ants of the Asian bark-nesting genus *Rhopalomastix* Forel, 1900 are a perfect case where a combined DNA-GM approach may be necessary for species discrimination. Workers of different species of *Rhopalomastix* can seem very similar in morphology, with no explicit features clearly separating congeners (*Wang, Yong & Jaitrong, 2018*, *2021*). Among these, the diagnosis and treatment of the species *R. javana* Wheeler, 1929 is particularly problematic. Based on past comparisons of mitochondrial DNA barcodes (COI), *R. javana* may actually comprise multiple closely-related species that genetically diverge from each other by uncorrected pairwise distances of >4% (see *Wang, Yong & Jaitrong (2021)*). However, because no definitive morphological differences could be recognized, these putative molecular species have been tentatively treated as a single species.

Currently, species of the *R. javana* complex can be diagnosed by the following characters in the worker caste, on top of traits defining the *R. rothneyi* species group (see *Wang, Yong & Jaitrong (2021)*): (1) Head slightly longer than wide, (2) propodeal junction in lateral view distinct and roundly obtuse, (3) propodeal dorsum differentiated from posterior declivity by weakly marginate angular edge, (4) posterolateral corners of propodeal dorsum angulate, (5) posterior propodeal face weakly marginate, (6) anterior clypeal face weakly and broadly convex in lateral view.

While a combined DNA-GM approach may have worked for *T. caespitum* (see *Wagner et al. (2017)*), the same results may not necessarily extrapolate to other species. The workflow must be applied and evaluated across different ant groups for a better sense of its broad utility. In this study, we therefore aim to validate and also test the reliability of a DNA-GM workflow in assessing cryptic diversity in ants, with focus on the *R. javana* species complex. Based on collective analyses of COI barcodes, genome-wide DNA and GM data, we want to identify cryptic species and draw insights on species boundaries within the complex. In the process, we will identify both advantages and possible drawbacks of an integrative DNA-GM pipeline on inferring cryptic diversity for ants in general. Through the use of GM, we also aim to flag external characters that may aid species discrimination not only within the *R. javana* complex, but for *Rhopalomastix* as a whole. This study represents a preamble to more in-depth taxonomic work on the *R. javana* species complex in future, which will build on results established therein.

**Table 1 Locality data and zoological reference collection (ZRC) codes associated with *Rhopalomastix* colony samples.**

| | Sample ID | General location | Specific locality | ZRC code |
|---|---|---|---|---|
| 1 | WW01 | Singapore | Mandai forest | ZRC_ENT00007584 |
| 2 | WW02 | North Thailand | Chiang Rai | ZRC_ENT00007609 |
| 3 | WW03 | Central Thailand | Nakhon Nayok Province, Khao Yai, 850 m | ZRC_ENT00007610 |
| 4 | WW04 | Northeast Thailand | Si Sa Ket Province, Phanom Dong Rak, Krabau Krabai | ZRC_ENT00007611 |
| 5 | WW05 | North Thailand | Nan Province, Nan Fa Resort | ZRC_ENT00007869 |
| 6 | WW06-07 | North Thailand | Nan Province, Pua District | ZRC_ENT00007870 |
| 7 | WW08 | North Thailand | Nan Province, Pua District, Sirapach Waterfall | ZRC_ENT00007871 |
| 8 | WW10 | North Thailand | Chiang Rai, Muang District, Khun Korn waterfall, >800 m | ZRC_ENT00007884 |
| 9 | WW16-17 | South Thailand | Trang Province, Na Yong District, Na Khao Sia Subdistrict, 28 m | ZRC_ENT00000942 |
| 10 | WW20-21 | Singapore | Mandai Track | ZRC_HYM0000576 |
| 11 | WW22-23 | Central Thailand | Nakhon Nayok Province, Muaeng District, Ban Hin Tang Subdistrict, 950 m | ZRC_ENT00013954 |
| 12 | WW25 | Peninsular Malaysia | Johor | ZRC_ENT00000750 |
| 13 | WW26 | Central Thailand | Nakhon Ratchasima Province, Pak Chong District, Mu Si, 800 m | ZRC_ENT00013953 |
| 14 | WW27-28 | Singapore | Upper Thomson Nature Park | ZRC_HYM0000287 |
| 15 | WW29 | Singapore | Mandai Road | ZRC_HYM0000508 |
| 16 | WW30-31 | South Thailand | Surat Thani Province, Tha Chang District, Tha Chang Subdistrict | ZRC_ENT00014143 |
| 17 | WW32 | South Thailand | Surat Thani Province, Tha Chang District, Tha Chang Subdistrict | ZRC_ENT00014144 |
| 18 | WW33 | South Thailand | Surat Thani Province, Tha Chang District, Tha Chang Subdistrict | ZRC_ENT00014145 |
| 19 | WW34 | South Thailand | Surat Thani Province, Tha Chang District, Tha Chang Subdistrict | ZRC_ENT00014146 |
| 20 | WW35 | West Thailand | Prachuap Khirikhan Province, Thap Sakae District, Huai Yang Subdistrict | ZRC_ENT00014149 |
| 21 | WW36 | South Thailand | Surat Thani Province, Tha Chang District, Tha Chang Subdistrict | ZRC_ENT00014150 |
| 22 | WW37 | Central Thailand | Saraburi province, Khaeng Khoi District, Cha-Om | ZRC_ENT00028483 |
| 23 | WW38-39 | Central Thailand | Saraburi province, Khaeng Khoi District, Cha-Om | ZRC_ENT00028484 |
| 24 | WW40-41 | Central Thailand | Saraburi province, Khaeng Khoi District, Cha-Om | ZRC_ENT00028485 |
| 25 | WW42-43 | Central Thailand | Saraburi province, Khaeng Khoi District, Cha-Om | ZRC_ENT00028486 |
| 26 | WW44 | Central Thailand | Nakhon Nayok Province, Muaeng District, Sarika Subdistrict, Khlong Maduea | ZRC_ENT00013974 |
| 27 | WW47-48 | Central Thailand | Nakhon Nayok Province, Muaeng District, Ban Hin Tang Subdistrict, Khao Khieao, 850 m | ZRC_ENT00013955 |
| 28 | WW49-50 | West Thailand | Prachuap Khirikhan Province, Thap Sakae District, Huai Yang Subdistrict | ZRC_ENT00014147 |

## MATERIALS AND METHODS

### Sample selection

For this study, 28 colony samples were selected from the Zoological Reference Collection (ZRC) at the Lee Kong Chian Natural History Museum (Table 1). These include three colonies that were morphologically identified as: *R. glabricephala, R. murphyi*, and an unnamed species of the *R. murphyi* group, respectively. The three samples were included as conceptual references, and as outgroup species for phylogenetic analyses. The other 25 colonies were morphologically identified as members of the *R. javana* complex, based on the characters listed in the Introduction. Each colony sample was allocated a short

numerical ID with prefix 'WW' for easier reference in downstream applications. Colony IDs comprising two numbers separated by a dash, correspond to the individual IDs of specimens from the same colony involved in downstream genome-wide sequencing after barcoding. For example, a colony sample with ID 'WW47-48' indicates that two specimens from the same colony with individual IDs WW47 and WW48 respectively, were used for genome-wide sequencing. Barcoded specimens were designated individual identifiers comprising the colony ID followed by an alphabet, *e.g.*, 'WW47-48-A'. Only individuals of the worker caste were used in this study.

## DNA barcoding

DNA extraction was performed on 1–8 representative specimens from each colony sample using QuickExtract™ DNA extraction solution (*Kranzfelder, Ekrem & Stur, 2016*), according to the manufacturer's instructions. Extracts were used as templates for subsequent COI (313 bp) barcoding. Amplification *via* polymerase chain reactions (PCRs) were performed following procedures described in *Wang et al. (2018)*, and with 13 bp-tagged forward and reverse primers designed for multiplexing of amplicons in subsequent MinION-based nanopore sequencing (*Srivathsan et al., 2019*). MinION reads were processed and barcodes were generated using the software ONTBarcoder (*Srivathsan et al., 2021*) following recommendations in *Srivathsan et al. (2021)*.

Objective clustering of successful barcodes generated by ONTBarcoder was performed according to protocols described in *Wang et al. (2018)*. In objective clustering, sequences are grouped according to uncorrected pairwise or p-distances based on the 'best close match' criteria (*Meier et al., 2006*; *Meier, Zhang & Ali, 2008*). Under this criteria, members of a set of putative conspecific sequences match one or more sequences in that set, within a defined percentage distance threshold. Cluster splitting and/or merging amongst barcodes were visualized as a dendrogram using the same software for objective clustering.

## Next generation sequencing (MIG-seq)

Reduced-representation genome sequencing by multiplexed inter-simple sequence repeats (ISSR) genotyping (MIG-seq) was performed, following the original protocols and primer sets detailed in *Suyama & Matsuki (2015)*, modified slightly from steps described in *Nguyen et al. (2020)*. About 2 ng of total genomic DNA per sample was used for the first PCR as template. Anonymous regions of ISSRs in the nuclear genome were amplified using Multiplex PCR Assay Kit Ver. 2 (Takara Bio, Shiga, Japan), with 7 µl reaction volumes. The thermal regime comprised one cycle of 1 min at 94 °C; 28 cycles of 30 s at 94 °C, 1 min at 38 °C, 1 min at 72 °C; and a final cycle of 10 min at 72 °C. Products from the first PCR were each diluted 25× with deionised water, before proceeding to the second PCR step. The second PCR was performed to add individual indices to each sample using common forward primers and indexed reverse primers (*Suyama & Matsuki, 2015*), with the following cycling conditions: 15 cycles of 10 s at 98 °C, 15 s at 54 °C, 1 min at 68 °C (10 µl reaction volumes). The second PCR was conducted using PrimeSTAR GXL DNA polymerase (Takara Bio, Shiga, Japan). Products from the second PCR were diluted 3× with deionised water and quantified for DNA concentration using a MultiNA Microchip
Electrophoresis System (Shimadzu, Kyoto, Japan). The diluted products were then normalized and pooled into a single library with nearly equimolar concentrations.

The pooled library was purified using QIAquick PCR Purification Kit (QIAGEN, Hilden, Germany). The purified library was run on a 2% agarose gel (DNA electrophoresis), and fragments of size range of 400–800 bp were isolated using GEL/PCR Purification Mini Kit (FAVORGEN Biotech, Pingtung, Taiwan). The DNA concentration of the size-selected library was measured using a SYBR green quantitative PCR assay (Library Quantification Kit; Takara Bio, Kusatsu, Japan; Clontech, Mountain View, CA, USA) with primers specific to the Illumina system. Roughly 4 nM of the concentration-adjusted library was denatured using 0.2 N NaOH, and mixed with Illumina-generated PhiX Control libraries (PhiX Control v3, Illumina, CA, USA) according to Illumina's recommendations. Finally, about 12 pM of library was used for paired-end sequencing on an Illumina MiSeq Sequencer using a MiSeq Reagent Kit v3 (150 cycles, Illumina); 80 bp of sequences were determined for read 1 and read 2, respectively. Raw reads from each indexed sample were demultiplexed using the index reads option of the sequencer.

A total of 11,645,626 raw sequence reads (232,912 per sample on average) was obtained from 50 individual specimen samples. However, the data quality of 11 samples was too poor to use for downstream analysis (these samples yielded only a few loci (3–59) in preliminary *de novo* assembly analysis). One sample was also omitted because initial quality checks revealed potential contamination and questionable data reliability. Thus, raw reads from just 38 individual samples were passed on for further processing and downstream analyses. Demultiplexed raw sequence reads (.fastq file) of the MIG-seq dataset are available online: https://doi.org/10.5281/zenodo.8177513.

### *De novo* assembly of MIG-seq dataset

The raw reads were processed by the sequence adapter trimming tool Trimmomatic 0.39 (*Bolger, Lohse & Usadel, 2014*) to remove reads containing adapter sequences for both the 5′ end (GTCAGATCGGAAGAGCACACGTCTGAACTCCAGTCAC) and the 3′ end (CAGAGATCGGAAGAGCGTCGTGTAGGGAAAGA) (parameters of ILLUMINACLIP and MINLEN options of Trimmomatic were set to "2:30:10" and "80" respectively). Reads 1 and 2 were pooled for each sample as independent reads (*i.e.*, not merged as paired-end reads).

Subsequent quality trimming and filtering of reads and *de novo* assembly of loci were performed using the pipeline as prescribed on ipyrad v0.9.8.2 (*Eaton & Overcast, 2020*). Quality trimming and filtering of the reads, followed by assembly and filtering of loci, were done using default settings for most parameters besides those otherwise mentioned here. Reads <75 bp in length after trimming were discarded (filter_min_trim_len = 75, filter_adapters = 1), and the datatype parameter was set to one of the single-end datatypes "ddrad". Minimum depth for majority-rule base calling (mindepth_majrule) was reduced from the default 6 to 3 to maximize available loci in each sample. The sequence similarity clustering threshold (clust_threshold) for the final dataset used in this study was 90%.

A concatenated supermatrix of MIG-seq loci was generated by ipyrad, excluding loci missing in >90% of samples (min_samples_locus = 4). More stringent filtering for missing data was not considered because it can be detrimental to the downstream phylogenetic analysis (*Eaton et al., 2017*). The final concatenated length of this aligned dataset was 567,960 bp, comprising 7,045 loci, 18,279 single-nucleotide polymorphisms (SNPs), and 4,785 parsimony-informative sites—this was used for phylogenetic analyses.

## Phylogenetic analyses

Concatenation-based phylogenetic analyses based on the MIG-seq dataset were conducted using both maximum likelihood (ML) and Bayesian inference (BI) approaches. Maximum likelihood analysis was performed using IQ-TREE 2.1.3, (*Minh et al., 2020*), with tests of branch supports by ultrafast bootstrap approximation (UFBoot; *Hoang et al., 2018*) and SH-approximate likelihood-ratio tests (SH-aLRT; *Guindon et al., 2010*), each with 1,000 replicates. The -bnni option was used to reduce the risk of overestimating the branch supports due to model violations. The GTR+I+R4 was used as the substitution model—this was selected based on BIC score comparison of different possible combinations of rate heterogeneity models (+I, +G, +I+G, +R, or +I+R) under the standard GTR model using ModelFinder (*Kalyaanamoorthy et al., 2017*) implemented in IQ-TREE.

Bayesian inference (BI) analysis under the GTR+G model was performed on the dataset using ExaBayes 1.5.1 (*Aberer, Kobert & Stamatakis, 2014*) on CIPRES Science Gateway (*Miller, Pfeiffer & Schwartz, 2010*: https://www.phylo.org/). Two independent simultaneous Markov chain Monte Carlo (MCMC) sampling runs were performed, each with two Metropolis-coupled chains and a sampling frequency of 500 generations. Parsimony trees were used as starting trees instead of random trees. The first 25% of generations was ignored as burn-in, and MCMC sampling was stopped when the two runs converged with <1% of the average standard deviation of split frequencies (ASDSF). The runs converged after 2,990,000 generations, and the effective sampling size (ESS) of every parameter was confirmed to be enough high (>1,000) using Tracer 1.7.1 (*Rambaut et al., 2018*).

## Population structure analyses

We further investigated underlying population structure in the *R. javana* complex based on the MIG-seq dataset, using fineRADstructure (*Malinsky et al., 2018*; version 0.3.1). The programme fineRADstructure—including RADpainter—is modified from fineSTRUCTURE (*Lawson et al., 2012*; *Lawson & Falush, 2012*) to allow the latter's application to RAD-seq-like datasets. It uses haplotype information of assembled loci to infer a "co-ancestry matrix"—a summary of nearest-neighbour haplotype relationships in the dataset—and can capture population structure with high resolution (*Lawson & Falush, 2012*; *Malinsky et al., 2018*).

The input files for RADpainter were converted using fineRADstructure-tool (https://github.com/edgardomortiz/fineRADstructure-tools) from the reassembled ipyrad outputs (.allele output obtained by adding "a" option to the output_formats parameter), excluding loci missing in >80% of samples. A final dataset of 35 ingroup samples comprising

3,294 loci was retained for the downstream fineRADstructure analysis.
The fineRADstructure clustering analysis, conducted on the co-ancestry matrix obtained with RADpainter, was run for 1,000,000 MCMC generations, with the first 50% of burn-in generations and sampling frequency of 1,000 generations (other settings were left on default). A dendrogram illustrating relationships of inferred populations was constructed using a built-in tool with 100,000 hill-climbing iterations.

## Geometric morphometrics
### Image acquisition
Three to five specimens from each successfully-sequenced colony sample (see Table 1) were imaged with a Dunn Inc.™ Passport II macrophotography imaging system, using a Mitutoyo 7D MKII objective lens with 10× zoom. Focus-stacked images were produced using Zerene Stacker (Zerene Systems LLC, Richland, WA, USA, http://zerenesystems. com/cms/stacker). Final images were annotated, and scale bars added using Adobe ® Photoshop 2020.

Each specimen was imaged in three standard perspectives used for visual examination in ant taxonomy, corresponding to the following anatomical aspects: (1) Head in full face view (hereafter referred to as 'Head'), (2) mesosoma in dorsal view (hereafter referred to as 'Meso'), and (3) mesosoma in profile view (hereafter referred to as 'Profile'). Three sets of images were thus obtained from 131 individuals, including 13 specimens from the three phylogenetic outgroup species. One Head image of a specimen identified as jsp1 was omitted from subsequent procedures as the specimen's head was visibly slightly damaged. A total of 392 images were then passed for landmark digitization.

## Landmark configuration and digitization
Landmarks are points of correspondence located at precise points over a structure, and can be matched among all specimens included in a study (*Manacorda & Asurmendi, 2018*). Comparisons of shape are typically performed between matching sets of landmarks—also known as 'configurations'—but not between individual landmarks. In this study, digitization of landmarks includes semilandmarks—points arranged along an outline that prescribe information on curvature (*Webster & Sheets, 2010*).

To ensure consistency and minimize variation in measurement, landmark digitization was wholly conducted by the first author. We first created a TPS text file—which lists image file names and stores their corresponding digitized two-dimensional landmark coordinates—from each directory containing image files for the respective perspectives, using tpsUtil v.1.78 (*Rohlf, 2015*). The TPS file for each perspective was then used to load and visualize the images in tpsDig2 v.2.31 (*Rohlf, 2015*). Using the same software, landmarks were subsequently digitized in the same order on every image, after setting a scale factor based on the scale bars added during image acquisition. A copy of each image was generated and digitized in the same manner, producing a replicate configuration for every image. These replicates were necessary for later assessment of measurement error.

The configuration for each Head image comprised 13 landmarks and 20 semilandmarks (two curves) (Fig. 1). For each Meso image, four landmarks and 38 semilandmarks

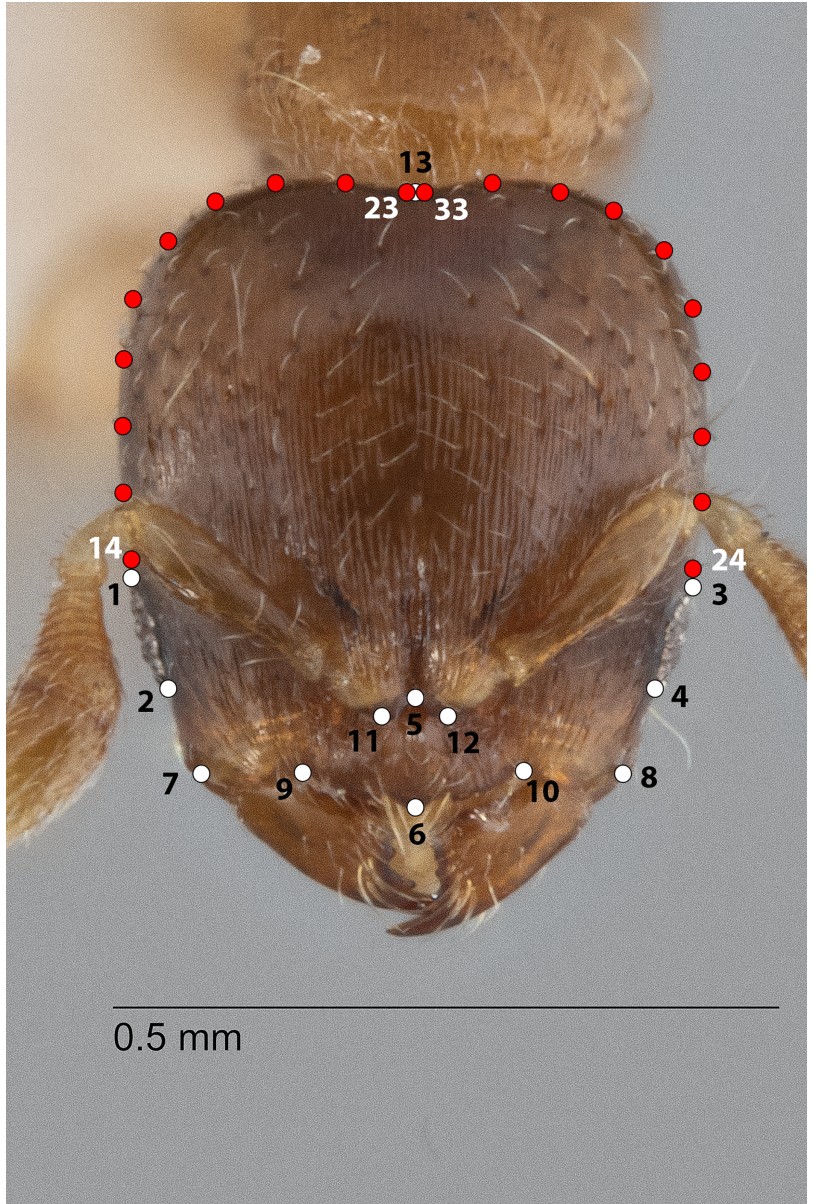

**Figure 1 Landmarks (white) and semi-landmarks (red) for Head aspect.**

(three curves) were digitized (Fig. 2). Finally, the configuration for each Profile image consisted of seven landmarks and 13 semilandmarks (one curve) (Fig. 3). Descriptions of landmarks and semilandmark positions for each image perspective are given in Tables S1A–S1C. Datasets of configurations for the three perspectives—Head (260), Meso (262), Profile (262)—were processed and investigated separately in downstream shape analyses. A breakdown of the numbers of individual configurations per species for each anatomical aspect is provided in Table 2.

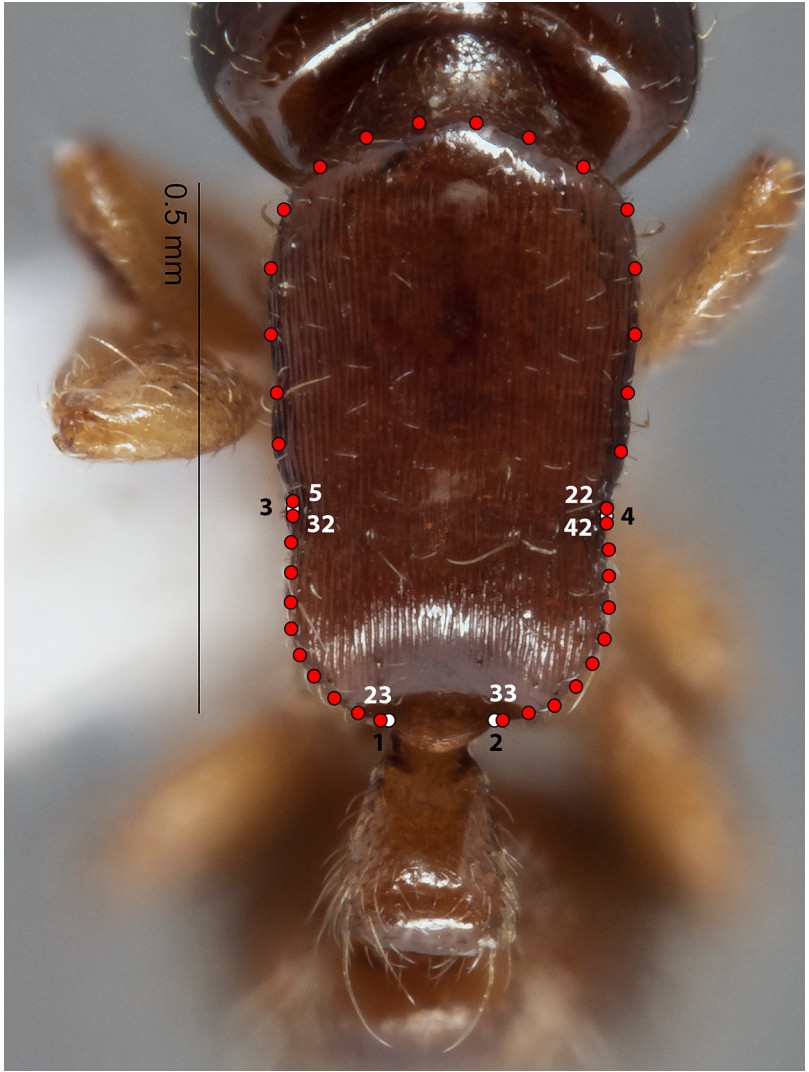

**Figure 2 Landmarks (white) and semi-landmarks (red) for Meso aspect.**

## Procrustes superimposition

Procrustes superimposition is a data transformation procedure involving the translation and rotation of landmark configurations into a common position, removing differences in orientation and size in the process. The shape configurations are projected from a non-linear shape space onto a tangent Euclidean plane. These projected shapes are then arranged and rescaled to minimize their average distances between the reference configuration—usually the calculated 'mean' shape from the dataset—and amongst each other. The resultant landmark coordinates are deemed 'Procrustes-aligned'.

We performed Procrustes superimposition for each perspective dataset, *i.e.*, Head, Meso, Profile, using tpsRelw v.1.70 (*Rohlf, 2019*). A special procedure was implemented for superimposition of semilandmark coordinates, with the use of a 'Sliders' file generated with tpsUtil v.1.78 (*Rohlf, 2015*). The software tpsRelw, using information provided in the

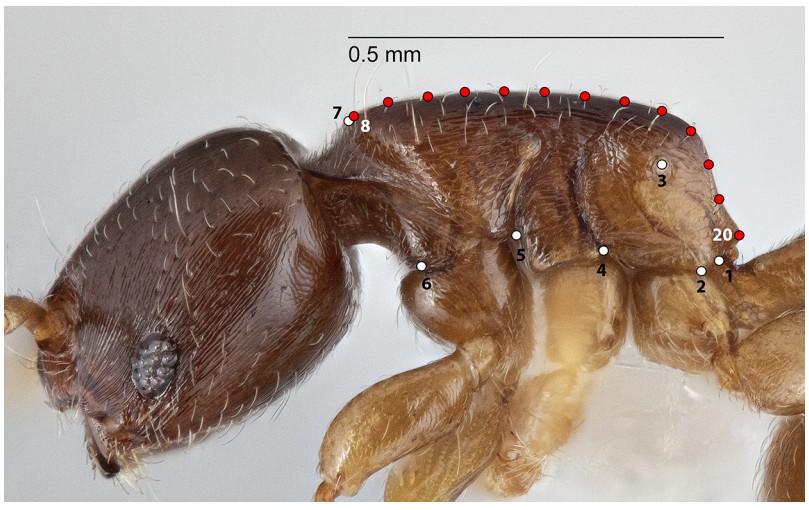

**Figure 3 Landmarks (white) and semi-landmarks (red) for Profile aspect.**

**Table 2 Breakdown of numbers of individual configurations/shapes per species analyzed in geometric morphometrics, for three anatomical aspects/perspectives.**

| Species | Anatomical aspect | | |
|---|---|---|---|
| | Head | Meso | Profile |
| jsp1 | 8 | 10 | 10 |
| jsp2 | 6 | 6 | 6 |
| jsp3 | 46 | 46 | 46 |
| jsp4 | 40 | 40 | 40 |
| jsp5 | 134 | 134 | 134 |
| *glabricephala* | 6 | 6 | 6 |
| *murphyi* | 10 | 10 | 10 |
| unk_outgroup | 10 | 10 | 10 |
| Total | 260 | 262 | 262 |

Sliders file, computationally slides semilandmarks along their constituent curves. The semilandmarks are slid until they are optimally-spaced from each other, whereby their spacing has minimal impact on overall differences between shapes (*Webster & Sheets, 2010*).

Upon completion of Procrustes superimposition, two types of data matrices were generated per perspective for downstream analyses: (1) Procrustes-aligned landmark coordinates and (2) relative warp scores. Relative warp scores are transformed linear combinations of the original landmark coordinates. Each warp score quantifies the contribution of a specific deformation to overall differences between reference and target shapes (*Webster & Sheets, 2010*).

For each anatomical perspective, two additional data subsets of Procrustes coordinates and relative warp scores respectively, were generated from only specimens of the *javana*

species complex (outgroup species omitted). The *javana* complex data subsets were analysed separately from total species datasets downstream.

Before proceeding to downstream analyses, we examined whether the original shape data was well-represented by the linear approximation. To do this, we computed the least-squares regression slope and correlation between the tangent (Euclidean) distances (to the reference shape) and original 'Procrustes distances' in non-linear shape space, for full datasets (*i.e.*, all species) of each anatomical aspect. Computations were performed using tpsSmall v.1.36 (*Rohlf, 2015*).

## Procrustes coordinates analysis

Procrustes-aligned coordinates were first analysed using the geomorph package (v4.0.5; *Baken et al., 2021*; *Adams et al., 2022*), on the R Statistical Software platform (v4.2.2; *R Core Team, 2022*). Permutation-based Procrustes ANOVA—also known as Goodall's F Test (*Goodall, 1991*)—was performed on all Procrustes-aligned datasets including those of only *javana* complex species per perspective, to assess potential effects of species, site, and replication (measurement error) on variation in inter-shape distances.

To visualize and roughly assess shape variation for each perspective, we performed between-group principal components analysis (PCA) (*Boulesteix, 2004*) on each dataset of Procrustes coordinates. In PCA, the Procrustes coordinates are transformed to form new truncated sets of independent variables which are linear combinations of the original variables—these are known as principal components (PCs). Principal component scores obtained from each PCA were plotted on a 2-D plane formed with the top two PCs as X (horizontal) and Y (vertical) axes. For between-group PCA, differences between specified group means are analysed regardless of within-group variation or differences among individuals. In this study, 'group' refers to the lineages or species supported by prior phylogenetic analysis, *i.e.*, *javana* complex species 1–5 and the three outgroup species.

Thin-plate spline (TPS) deformation grids depicting bending energy changes in overall structures—Head, Meso and Profile—were generated for each PCA, to visualize possible trajectories of shape changes between species. These deformation grids were also incorporated with Jacobian expansion factors, which results in differential-colouration of grid areas according to the extent of 'shrinkage' or 'expansion' from the hypothetical reference shape.

Principal components analyses were performed, and TPS deformation grids were generated using PAST v.4.06b (*Hammer, Harper & Ryan, 2001*).

## Relative warps analysis

Canonical variates analysis (CVA) was conducted on each dataset of relative warp scores. The process of CVA is similar to PCA, unlike PCA however, canonical variates (CVs) are constrained to maximize separation between *a priori* groups (species), based on patterns of within-group variation. Canonical variate scores obtained from each round of CVA were plotted on a 2-D plane formed with the top two CVs as X and Y axes.

To assess how effectively specimens could be correctly assigned to their pre-defined groups (species) based on shape differences as prescribed *via* the CVs from each dataset—*i.e.*, classification accuracy—we performed a jack-knifed cross-validated specimen reassignment procedure. In this procedure, each individual is omitted from calculations before its group membership is calculated based on data from all remaining individuals. Linear discriminant classifiers—'classifiers' for short—calculated from CVA basically assign each individual in the dataset to the group that renders the individual closest to the group mean. A confusion matrix—which depicts the number of individuals in each pre-defined group (rows) that were designated different groups (columns) by the classifier—was constructed for each round of CVA.

Finally, to test the significance of shape differences between species, we conducted posthoc pairwise permutation-based MANOVA (PERMANOVA) on relative warp scores from the three perspectives, between all pairs of species. To investigate potential shape differences linked to location, for each putative species with samples from multiple geographic subregions, *i.e.*, different 'sites', we performed pairwise PERMANOVA tests between samples grouped by site. Significance of each test statistic—a pseudo ANOVA F-ratio, referred to hereafter as 'F-value' for brevity—was computed by permutation of group membership with 9,999 replicates. Resulting $p$ values were Bonferroni-corrected for multiple comparisons.

Relative warps analyses were carried out on PAST v.4.06b (*Hammer, Harper & Ryan, 2001*).

# RESULTS

## DNA barcoding and objective clustering

A total of 108 specimens from across 28 colonies—Singapore and Thailand in origin—were successfully barcoded for COI (GenBank accession numbers: OR262214–OR262321). Objective clustering of the sequences yielded eight molecular species units at ≥4% threshold distance, including the three recognized outgroup species (Fig. 4). The three outgroup species diverged from the *R. javana* complex specimens by ≥18.9% uncorrected p-distance. Five putative molecular species units—referred to as 'jsp1-jsp5' hereafter—were apparent in the *R. javana* complex at 4% threshold distance, with inter-unit divergences ranging from 4.2% to 10.6%. Two individuals from the same colony (WW26) in Central Thailand (CTH) were delimited from the rest of the complex by 10.6%—this was termed 'jsp1'. One individual from another CTH colony then split from the remaining group by 5.8%—this was named 'jsp2'.

Species clusters inferred for the rest of the specimens based on COI were not as straightforward. The putative species 'jsp3' was distinguished from the others by a narrow 4.5% threshold distance. The remaining species units—'jsp4' and 'jsp5', were divergent by a slightly lower 4.2%. The former putative species comprises specimens from North (NTH) or Northeast Thailand (NETH), as opposed to those of jsp5, which originate from other parts of Thailand and Singapore. Considering the possibility of jsp4 and jsp5 being allopatric clusters of the same species, we nevertheless retained jsp4 as a separate putative group for downstream comparisons, simply to err on the side of caution. If subsequent

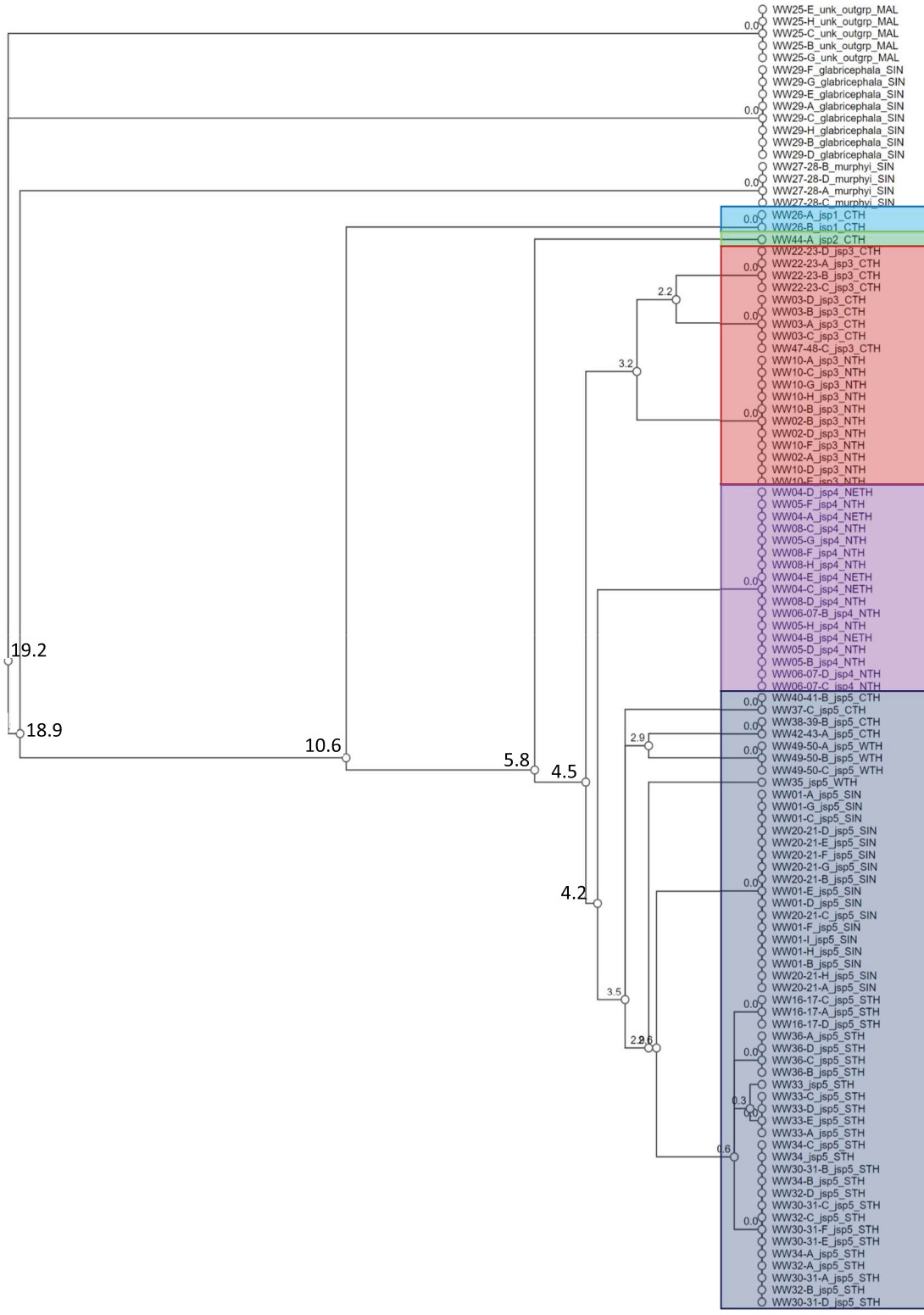

**Figure 4 Cluster dendrogram based on objective clustering of COI (313 bp).** Values at nodes indicate percentage thresholds of uncorrected pairwise distances (p distances) at which clusters or sequences diverge. Sequences belonging to the outgroup or same putative *R. javana* complex species cluster (jsp1-jsp5) are highlighted in colour.

shape analyses reveal no significant differences in morphology between jsp4 and jsp5, that may strengthen the hypothesis of allopatric conspecificity.

Within jsp3 and jsp5 respectively, there appeared to be further subgroups distinctly separated by geography, albeit by threshold distances of <4%. For jsp3, CTH and NTH colonies were split at 3.2% (Fig. 4); in jsp5, West Thailand (WTH) and CTH colonies diverged from Singapore (SIN) and South Thailand (STH) colonies by 3.5%.

### Phylogenetic analyses

Phylogenetic trees generated by both ML and BI methods shared similar topologies and were convergent in resolved clades. Both ML and BI trees recovered five major lineages belonging to the *R. javana* complex—in agreement with the putative species units (jsp1-jsp5) derived from objective clustering of COI. Most of the major lineages in the best ML tree were resolved with high confidence, supported by robust SH-aLRT and maximum ultra-fast bootstrap support values of 100% at the relevant branches (Fig. 5A). Only the relationship between jsp2—represented by sample WW44—and the clade of jsp3-5 had relatively weak support, with lower SH-aLRT and ultra-fast bootstrap values of 87% and 65% respectively.

In contrast, posterior probabilities for the major lineages of the *R. javana* complex in the BI consensus tree were all calculated as the maximum value of 1 (Fig. 5B). This included the relationship of jsp2 with the jsp3-5 clade, unlike in the best ML tree where jsp2 was sister to jsp3-5 with relatively weak support.

### Population structure and co-ancestry

Population clustering by fineRADstructure based on the co-ancestry matrix recovered distinct clusters for jsp3 and jsp4, while multiple subclusters were recovered within jsp3 and jsp5 (Fig. 6). The pattern of terminal population assignments was largely consistent with branching patterns in both the ML and BI phylogenetic trees. However, the two putative species represented by singletons—jsp1 and jsp2—were assigned to the same terminal population in the cluster dendrogram.

The co-ancestry matrix heatmap depicts relatively strong haplotype sharing—higher levels of co-ancestry—within each of jsp3 and jsp4, from other species of the *R. javana* complex (Fig. 6). In contrast, jsp5 appears to have weaker shared ancestry within itself. The putative species jsp1 and jsp2 also each appear to be genetically distinct from the rest, with low levels of shared ancestry from other clusters.

Multiple subclusters demonstrating significantly higher co-ancestry within each of clusters jsp3 and jsp5 appear associated with geographical location. Within jsp5, there seems to be slightly higher admixture between CTH and WTH populations, whereas STH populations are genetically closer to those from SIN. For jsp3, NTH and CTH populations are more distinctly separated from each other—shared ancestry within each population strongly exceeds that between the two groups.

## A. Maximum Likelihood (ML) – Best tree

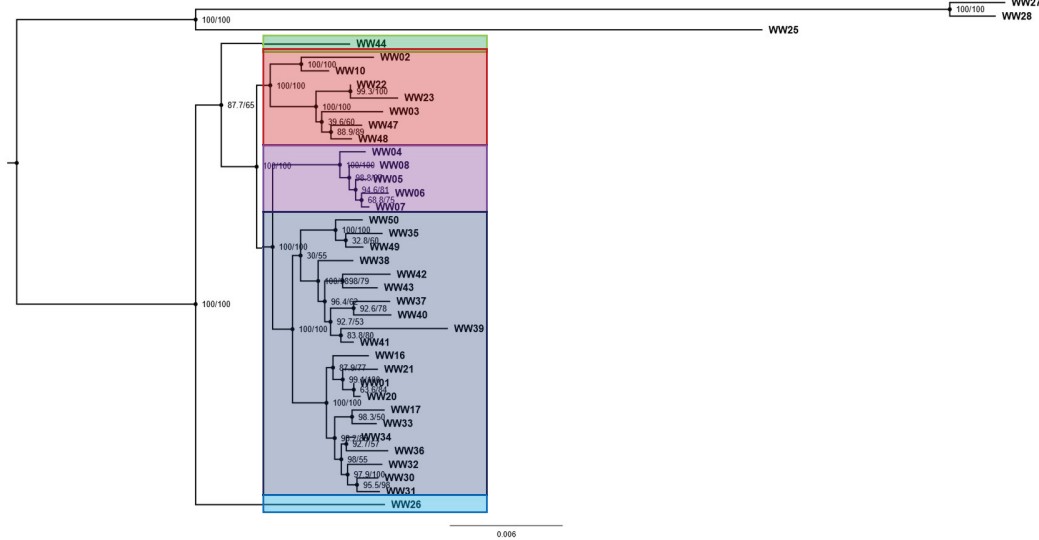

## B. Bayesian Inference (BI) – Consensus tree

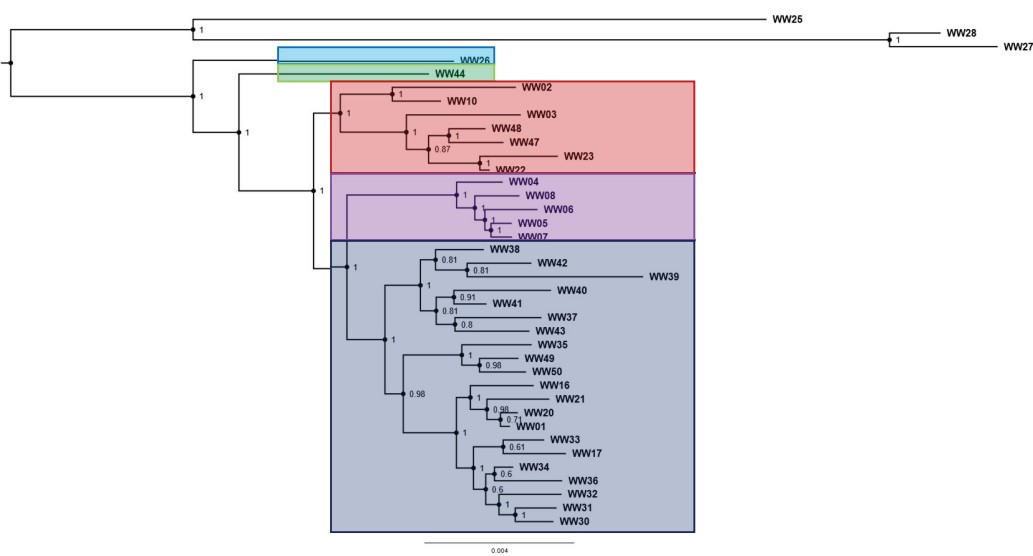

**Figure 5 Phylogenetic trees for *Rhopalomastix* based on concatenated MIG-seq data.** (A) Best tree generated based on maximum likelihood (ML). Values at the nodes comprise SH-aLRT (left) and ultrafast bootstrap (right) support values generated from 1,000 replicates each. (B) Consensus tree generated using the Bayesian Inference (BI) approach. Values at nodes indicate posterior probabilities. Outgroup species are highlighted in yellow.

## Geometric morphometrics analyses
### *Assessment of tangent space approximation*
Regressions of Euclidean distances in tangent space (Y) against original Procrustes distances in non-linear shape space (X) showed slopes of >0.99 and correlations close to

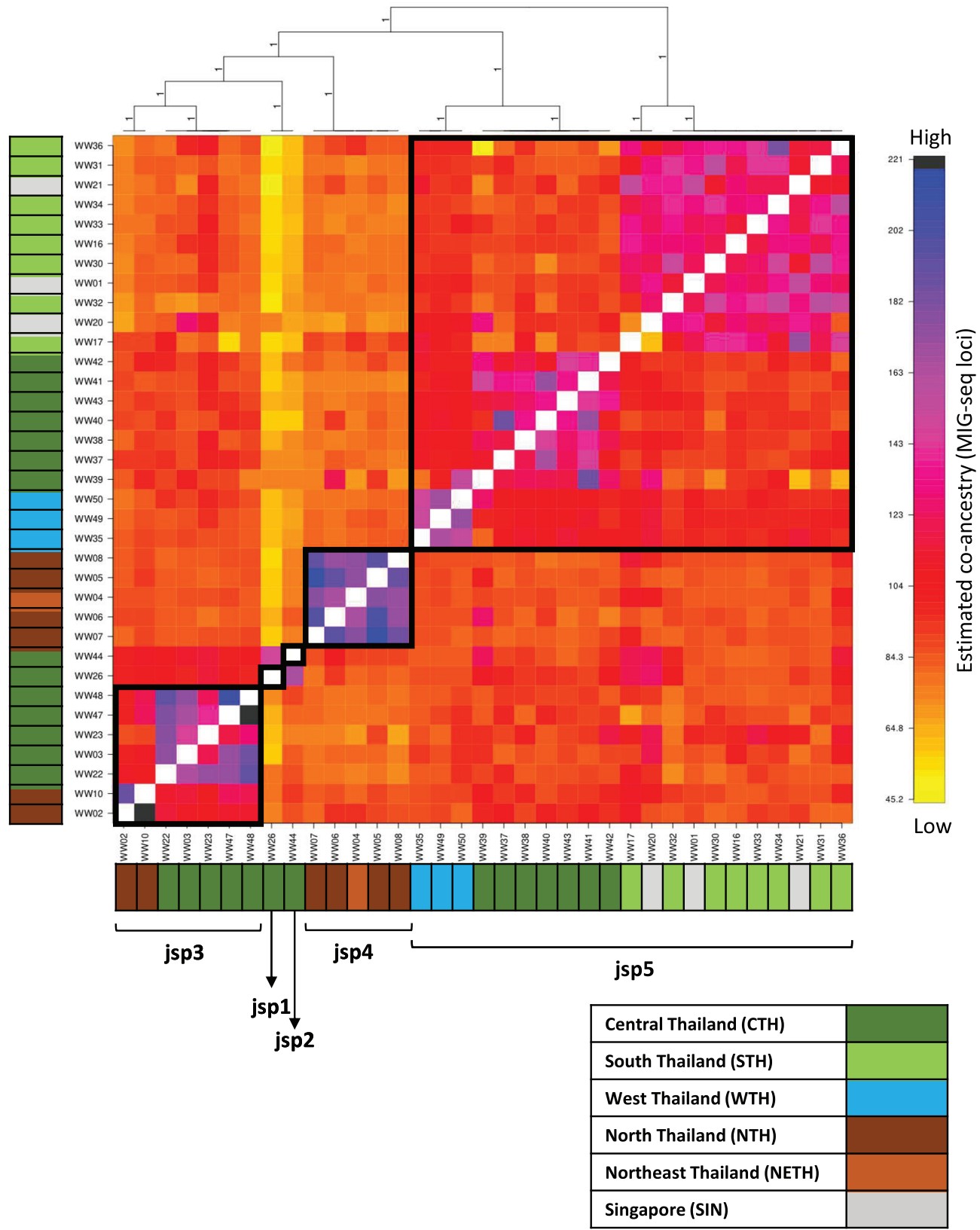

**Figure 6 Clustered fineRADstructure co-ancestry matrix estimated from MIG-seq loci.** Cluster dendrogram (top) and heatmap differentiating levels of estimated co-ancestry based on a gradient of colour tones (centre-main). Cluster support values are indicated as posterior probabilities on

**PeerJ** ________________________________________________

**Figure 6 (continued)**
corresponding branches of the dendrogram—in this case, all values = 1. Colour bars along the vertical and horizontal axes indicate geographic
origins of each sample, according to the legend on bottom-right.

1.0 for all anatomical perspectives (Fig. S1). These indicate that projections of shape onto linear shape space are good approximations or representations of the actual shapes under study.

### Procrustes ANOVA

Procrustes ANOVA revealed statistically significant effects ($p < 0.001$) of 'species' and 'site (location)' on shape variation between all species, within each of the three imaged anatomical perspectives (Table S2). For Head shape, 'species' explained 28.9% of variance —reflected by the Total Sum of Squares (TSS)—while 'site' accounted for 6.2% variance. 'Species' explained 21%, and 'site' explained 6.3% of TSS for Meso shape. For Profile shape, 'species' accounted for just 10.6% of variation, while 'site' explained 8.6%.

Relative to comparisons among all species, 'species' explained less of variation in Head (14.4%) and Meso (12.1%) shapes among species of the *R. javana* complex. The same factor, however, accounted for an identical proportion of variance (10.6%) in Profile shape. Unlike 'species', 'site' explained slightly more variation (compared to all-species comparisons) in Head shape (8.2%) than Meso (7.6%) and Profile (7.7%) shapes. Both effects were considered statistically significant across all three perspectives ($p < 0.001$).

Replication—represented by the factor 'rep' included in the model—explained 0.09–0.2% of TSS for all species over the three perspectives. As these effects were statistically insignificant, *i.e.*, $p > 0.1$, we considered measurement error to be negligible.

### Procrustes PCA

For Head shape among all species, 98.6% of variation could be summarized by the top five principal components (PCs) of the PCA, with PC1 (54.5%) and PC2 (27.7%) accounting for most of observed variance (Fig. 7A). Thin-plate spline deformations along PC1 suggest that different species could be most variable in terms of the shape of the median clypeal disc, and the shortest distance of the eye from mandibular insertion in full face view (Fig. 7B).

As for Head shape among species of the *R. javana* complex (Fig. 7A), PC1 explained about 65% of variation. Changes along PC1 seemed to arise mainly from variation in the shape of the median clypeal disc, depth of the median concavity or depression on the posterior margin of head, and to a moderate extent, eye size (Fig. 7C). PC2 explained 20.7% of variation in Head shape, with changes along the axis mainly associated with variation in size and shape of the clypeal and malar regions (Fig. 7D).

For Meso shape among all species, 98.4% of variation could be summarized by the first five PCs of the PCA, with PC1 explaining 60.4% and PC2 17.4% of the variance respectively (Fig. 8A). No one feature of Meso shape appeared to stand out in terms of varying the most along PC1 (Fig. 8B). Heightened variation reflected at the inflection

Peerj

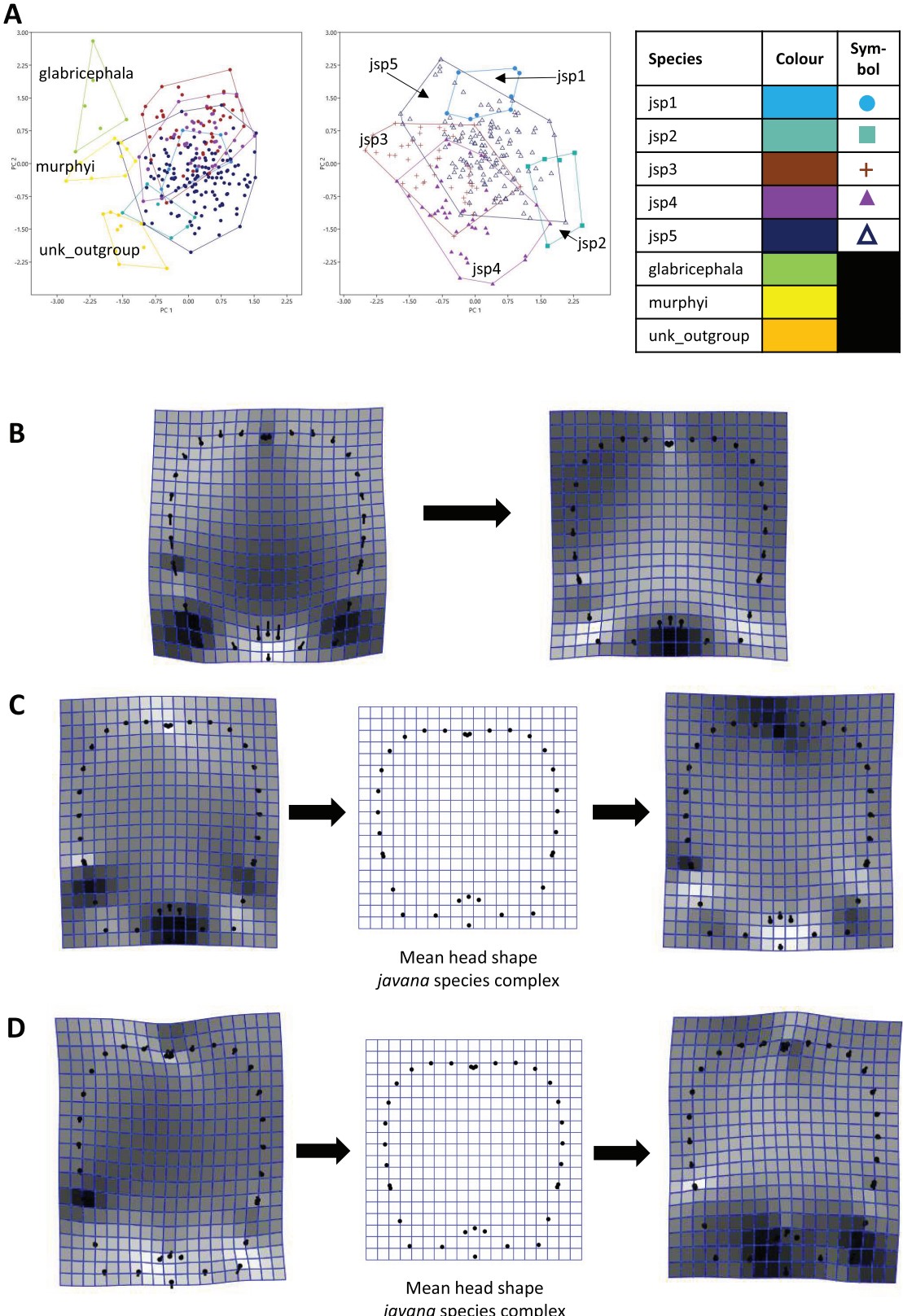

**Figure 7 Procrustes-aligned coordinates analyses for Head shape.** (A) Between-group PCA for Procrustes coordinates of all species (left), and jsp1-jsp5 (right) (PC2 *vs* PC1). PC axes are scaled to Eigenvalues for increased clarity. (B–D) Thin-plate spline (tps) deformation grids depicting

**Figure 7 (continued)**
changes in bending energy and shape variation. (B) Changes along PC1 for all species [−0.06 >> 0.03]; (C) changes along PC1 for *javana* complex species [−0.02 >> 0.02]; (D) changes along PC2 for *javana* complex species [−0.03 >> 0.03]. Colour tones represent variation in magnitudes of deformation in grid elements, quantified by Jacobian expansion factors—white/pale = contraction, black/dark = expansion.

points separating propodeum from mesonotum and petiolar peduncle, might be artefacts of minor measurement bias.

As with Head shape, variation in Meso shape amongst species of the *R. javana* complex could be explained mostly by PC1 (65.8%) (Fig. 8A). Deformation changes along PC1 mostly arose due to increased variation in curvature of the dorsolateral propodeal margins (Fig. 8C). Shape changes along PC2 (16.0%), in contrast, were associated with heightened variation in multiple traits: (1) general width of promesonotum, (2) condition of anterior margin of pronotal disc, (3) depth of inflections separating propodeum from mesonotum, and (4) dorsal width of peduncular articulation with propodeum (Fig. 8C).

Similarly for Profile shape, 97.9% of variation among all species could be summarized by the first five PCs of the PCA, with PCs 1 and 2 explaining 38.4% and 27% of variance respectively (Fig. 9A). Changes along PC1 appear linked to broad variation in pro-coxa size and longitudinal width of the lower half of pronotum in lateral view, and also variable curvature of the entire mesosomal dorsum outline (Fig. 9B).

Among species of the *R. javana* complex, variation in Profile shape could be explained mostly by PC1 (49.1%) and PC2 (41.7%) in near-equal measure (Fig. 9A). Changes along PC1 were largely associated with variable pro-coxa size—in particular its longitudinal width in profile view—and, to a lesser extent, the relative size and shape of the metapleural area below the propodeal spiracle (Fig. 9C). Changes along PC2 also seem closely linked to high variation in pro-coxa size, in addition to variable curvatures of the mesosomal dorsum and posterior propodeal declivity (Fig. 9D).

## CVA and classification accuracy

In the CVA of relative warps of Head shape, the top two canonical variates (CVs) combined explained 59.9% of differences among all species, with CVs 1 and 2 accounting for 34.1% and 25.8% of total variance respectively. For Head shape among *R. javana* complex species alone, the first two CVs explained a larger proportion of total variation— 76.5% with CVs 1 and 2 covering 48.9% and 27.6% separately. (Fig. 10A).

Jack-knifed classification accuracy (referred to simply as 'classification accuracy' hereafter) based on the CVA of Head shape for all species was quite high, with 80.8% of individuals correctly assigned to their original groups/species. Classification accuracy for *R. javana* complex species only was slightly lower, with 77.8% of individuals accurately designated to their a-priori groups/species (Table S3).

For relative warps of Meso shape, the first two CVs together accounted for 52.9% of differences amongst all species—28.8% and 24.1% explained by CVs 1 and 2 respectively. For *R. javana* complex species, the first two CVs explained 68.4% of total variance—38.7% and 29.7% covered by each of CVs 1 and 2 (Fig. 10B).

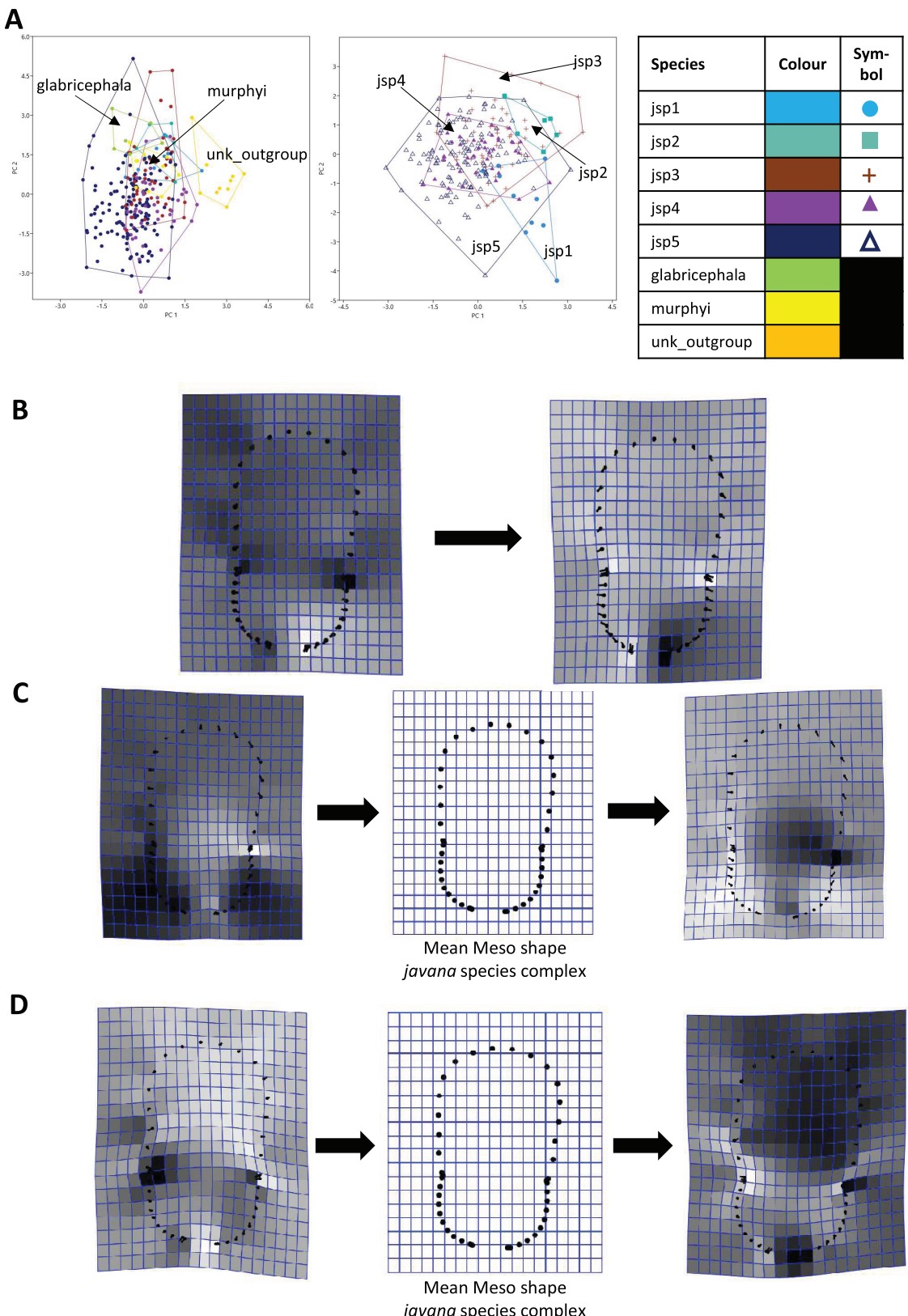

**Figure 8 Procrustes-aligned coordinates analyses for Meso shape.** (A) Between-group PCA for Procrustes coordinates of all species (left), and jsp1-jsp5 (right) (PC2 *vs* PC1). PC axes are transformed to Eigenvalues for increased clarity. (B–D) Thin-plate spline (tps) deformation grids

**Figure 8 (continued)**
depicting changes in bending energy and shape variation. (B) Changes along PC1 for all species [−0.04 >> 0.06]; (C) changes along PC1 for *javana* complex species [−0.04 >> 0.04]; (D) changes along PC2 for *javana* complex species [−0.03 >> 0.03]. Colour tones represent variation in magnitudes of deformation in grid elements, quantified by Jacobian expansion factors—white/pale = contraction, black/dark = expansion.

Classification accuracy based on the CVA of Meso shape for all species was lower relative to Head shape—only 60.7% of individuals were correctly assigned to their pre-defined groups/species. Classification accuracy of Meso shape for only *R. javana* complex species was no better, with only 60.6% of individuals correctly assigned (Table S4).

Finally, in the CVA of relative warps of Profile shape for all species, the top two CVs together accounted for 64.1% of total variation—45.1% and 19% explained by CVs 1 and 2 separately. Similarly for Profile shape among *R. javana* complex species, the first two CVs explained slightly more of total variation—69.9%—with CVs 1 and 2 covering 45.3% and 24.6% respectively (Fig. 10C).

Classification accuracy based on the CVA of Profile shape for all species was 74.4%—higher than that of Meso shape, but still slightly less than that of Head shape. Classification accuracy was almost identical for Profile shape amongst *R. javana* complex species, with 74.2% of individuals correctly re-assigned to their original groups/species (Table S5).

## Pairwise PERMANOVA

Pairwise PERMANOVA performed on relative warp scores of Head shape indicated significant (*i.e.*, $p < 0.05$ after Bonferroni correction; exact corrected p-values provided in Table S6) differences between most species, except for three pairs: jsp1-jsp5, jsp2-*R. glabricephala*, *R. murphyi-R. glabricephala* (Table 3A). In contrast, many more species pairs showed no significant differences (*i.e.*, $p \geq 0.05$) in Meso shape: jsp1-jsp2/jsp3/ *R. murphyi/R. glabricephala*, jsp2-jsp3/jsp4/jsp5/*R. murphyi/R. glabricephala*, jsp3-*R. murphyi/R. glabricephala*, *R. murphyi-R. glabricephala* (Table 3B). Profile shape was also not significantly different among many species pairs, though not as numerous as for Meso shape: jsp1-jsp2, jsp2-jsp3/jsp4/unk_outgroup/*R. glabricephala*, jsp4-jsp5/*R. glabricephala*, jsp5-*R. glabricephala*, *R. murphyi-R.glabricephala* (Table 3C). All putative species of the *R. javana* complex—jsp1-jsp5—were significantly different from each other in at least one anatomical aspect. Only two species pairs: jsp2 and *R. glabricephala*, and the outgroup species *R. murphyi* and *R. glabricephala*, were inferred as statistically indistinguishable in all three shape aspects.

PERMANOVA tests between Central Thai (CTH, 28 images) and North Thai (NTH, 18 images) populations of jsp3 revealed significant differences in Head (F = 3.097, $p = 0.0095$), Meso (F = 2.89, $p = 0.0068$), and Profile (F = 2.749, $p = 0.022$) shapes. For jsp5, pairwise tests between CTH (42 images), South Thai (STH, 54 images), West Thai (WTH, 18 images) and Singapore (SIN, 20 images), showed mostly significant differences (Bonferroni-corrected $p < 0.05$, exact $p$ values given in Table S7) across all three anatomical aspects. For both Head and Profile shapes, only WTH and STH populations of jsp5 were

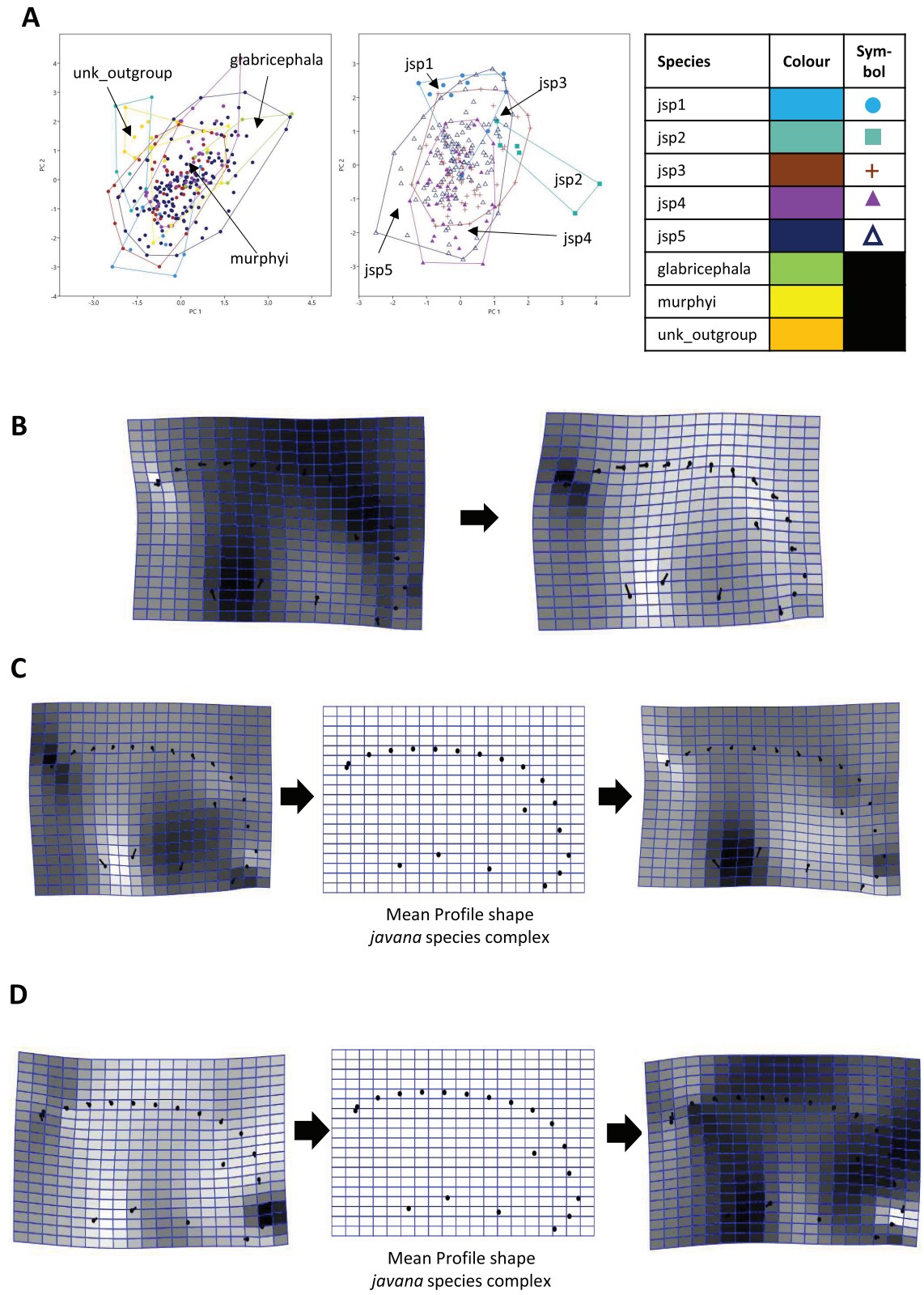

**Figure 9 Procrustes-aligned coordinates analyses for Profile shape.** (A) Between-group PCA for Procrustes coordinates of all species (left), and jsp1-jsp5 (right) (PC2 *vs* PC1). PC axes are transformed to Eigenvalues for increased clarity. (B–D) Thin-plate spline (tps) deformation grids

**Figure 9 (continued)**
depicting changes in bending energy and shape variation. (B) Changes along PC1 for all species [−0.04 >> 0.06]; (C) changes along PC1 for *javana* complex species [−0.04 >> 0.04]; (D) changes along PC2 for *javana* complex species [−0.03 >> 0.03]. Colour tones represent variation in magnitudes of deformation in grid elements, quantified by Jacobian expansion factors—white/pale = contraction, black/dark = expansion.

found to be not significantly different (Tables 4A, 3C). For Meso shape, only SIN and STH populations did not significantly differ from each other (Table 4B).

## DISCUSSION

Our results from genetic and geometric morphometric (GM) analyses collectively support the heterospecificity of the cryptic *R. javana* complex. Phylogenetic analyses based on genome-wide MIG-seq data recovered and strongly supported the monophyly of five lineages—jsp1-5—(Fig. 5); these lineages arecongruent with putative species clusters inferred from mitochondrial COI (Fig. 4). Co-ancestry analyses using fineRADstructure also provided additional support for strong genetic distinctiveness of sympatric and/or allopatric populations of the putative species. Subsequent GM analyses for three standard morphological aspects—Head, Meso and Profile—indicated significant differences between all pairs of putative *R. javana* species in at least one aspect (Table 3). These results shed crucial insights on cryptic diversity in *R. javana*, setting the stage and providing direction for further taxonomic investigation to formally establish the five species previously treated as conspecific.

Distinguishing intraspecific population genetic structure from interspecific boundaries is challenging—arguably, many methods of molecular species delimitation tend to confuse the two concepts (*Sukumaran & Knowles, 2017*; *Chan et al., 2020, 2022*). Deep genetic splits between allopatric populations may simply arise due to geographic isolation over time and genetic drift, however, this exclusivity does not guarantee that these lineages would stay distinct if they occur in sympatry (*Meier, Zhang & Ali, 2008*; *Harrison & Larson, 2014*). To better resolve species boundaries, it is necessary to understand the evolutionary processes and possible mechanisms underlying speciation in each focal taxonomic group. For the morphologically cryptic but closely-related species of the *R. javana* complex, the main conundrum we have to address is: how may species boundaries be established and maintained between genetically-differentiated lineages?

### Inferring species boundaries

The presence of well-supported lineages or clusters existing in broad sympatry, or near-sympatry, is one form of compelling evidence for multiple species occurrence, as it suggests implicit mechanisms driving reproductive isolation between populations existing in relatively close proximity. We observed this in our results, especially for colonies obtained from Central Thailand. Three genetically-distinct lineages (Figs. 4–6)—jsp1, jsp2 and jsp3—were recovered from colonies in the same general vicinity shared between Nakhon Ratchasima (jsp1) and Nakhon Nayok (jsp2, jsp3) provinces, while a fourth lineage—jsp5—was found in the adjacent Saraburi province (Fig. 11A). We observed a

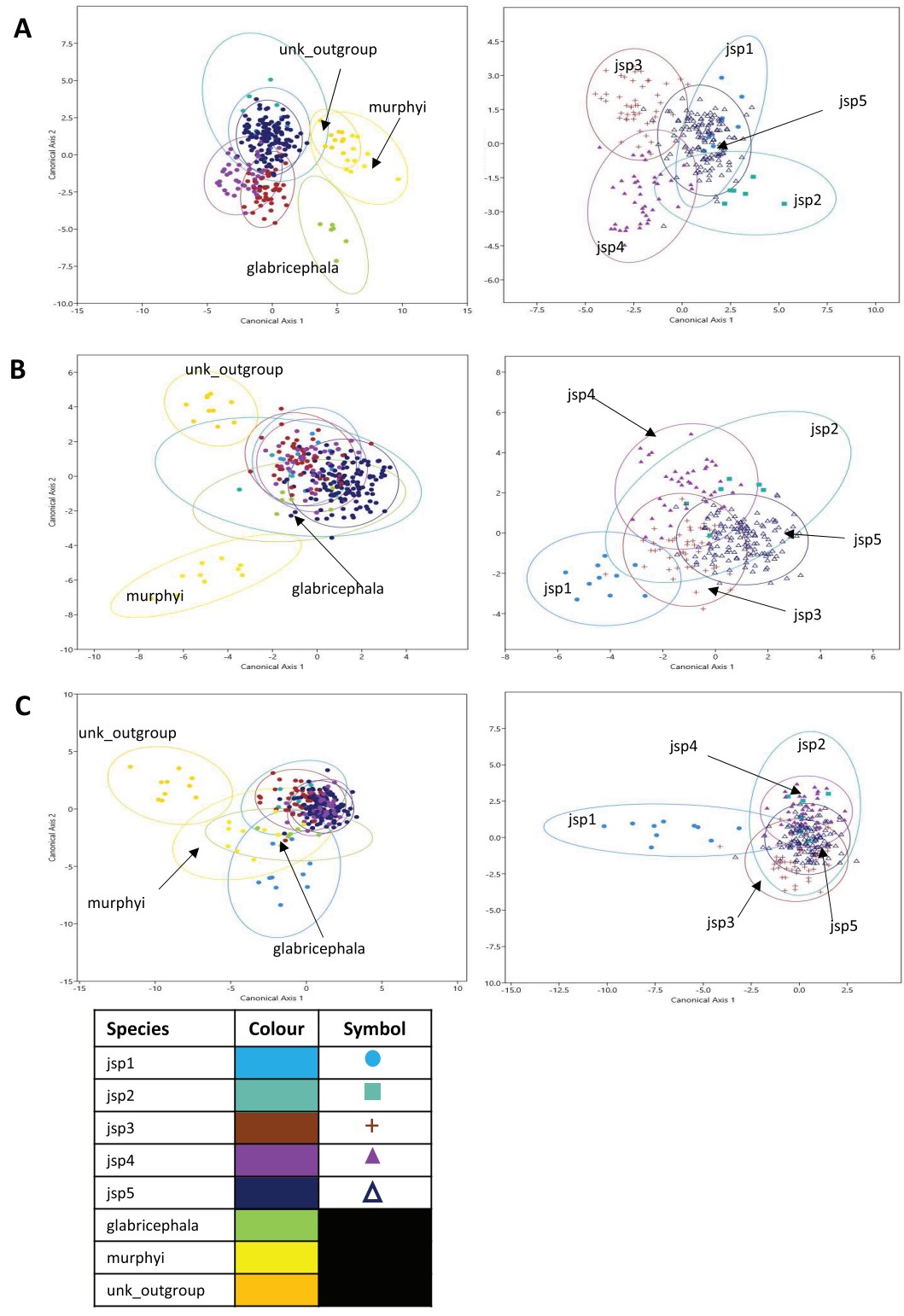

**Figure 10 Canonical variates analyses (CVA) of relative warps.** CVA plots (Y-axis—Canonical axis 2; X-axis—Canonical axis 1) for all species (left) and jsp1-jsp5 (right), based on relative warps from (A) Head, (B) Meso and (C) Profile shapes. Species clusters are encircled by 95% ellipses.

**Table 3 Pseudo-F values from pairwise PERMANOVA of relative warp scores from three anatomical aspects between *Rhopalomastix* species.**

| Species | jsp1 | jsp2 | jsp3 | jsp4 | jsp5 | unk_outgroup | *murphyi* | *glabricephala* |
|---|---|---|---|---|---|---|---|---|
| **A. HEAD** | | | | | | | | |
| jsp1 | 0 | | | | | | | |
| jsp2 | 4.087 | 0 | | | | | | |
| jsp3 | 5.03 | 11.09 | 0 | | | | | |
| jsp4 | 5.912 | 9.004 | 9.591 | 0 | | | | |
| jsp5 | **2.264** | 4.968 | 18.14 | 13.15 | 0 | | | |
| unk_outgroup | 11.37 | 5.659 | 27.73 | 30.52 | 18.91 | 0 | | |
| *murphyi* | 11.41 | 9.396 | 23.1 | 30.27 | 24.42 | 8.809 | 0 | |
| *glabricephala* | 7.574 | **6.532** | 15.57 | 17.95 | 20.76 | 8.705 | **3.079** | 0 |
| **B. MESO** | | | | | | | | |
| jsp1 | 0 | | | | | | | |
| jsp2 | **2.728** | 0 | | | | | | |
| jsp3 | **2.829** | **1.085** | 0 | | | | | |
| jsp4 | 4.841 | **1.535** | 4.727 | 0 | | | | |
| jsp5 | 8.129 | **2.963** | 14.68 | 9.337 | 0 | | | |
| unk_outgroup | 11.57 | 11.36 | 16.03 | 19.38 | 34.63 | 0 | | |
| *murphyi* | **3.016** | **2.285** | **2.862** | 4.723 | 6.114 | 14.59 | 0 | |
| *glabricephala* | **3.368** | **3.209** | **2.244** | 5.628 | 5.129 | 16.45 | **2.935** | 0 |
| **C. PROFILE** | | | | | | | | |
| jsp1 | 0 | | | | | | | |
| jsp2 | **3.29** | 0 | | | | | | |
| jsp3 | 5.167 | **3.34** | 0 | | | | | |
| jsp4 | 6.246 | **4.276** | 5.129 | 0 | | | | |
| jsp5 | 5.197 | 4.708 | 6.225 | **3.745** | 0 | | | |
| unk_outgroup | 7.969 | **3.108** | 4.325 | 7.173 | 8.568 | 0 | | |
| *murphyi* | 7.468 | 6.737 | 8.168 | 8.904 | 9.839 | 10.52 | 0 | |
| *glabricephala* | **1.387** | **3.565** | 4.977 | **4.718** | 3.785 | 8.985 | **4.221** | 0 |

**Note:**
Statistically insignificant F-values—Bonferroni-corrected $p$ values $\leq 0.05$—are indicated in bold. Exact $p$ values are provided in Table S6.

similar result with populations from North Thailand, where two discrete lineages jsp3 and jsp4 were identified from Chiang Rai and the neighbouring Nan Province respectively (Fig. 11B). The occurrences of these lineages in near-sympatry suffice to demonstrate that *R. javana* definitely comprises more than one species.

A key factor contributing to restriction of gene flow between (near-) sympatric lineages in Central and North Thailand may possibly be elevation and its influences on ant adaptation. The occurrence of each putative species appears limited to areas of either relatively low or high elevation, but never both: (1) 800–950 m—jsp1, jsp3; (2) <400 m— jsp2, jsp4, jsp5. Organisms typically face major physiological challenges at high elevation, where conditions are characterized by low atmospheric oxygen pressure or environmental hypoxia (*Dahlhoff et al., 2019*), low annual minimum and mean temperatures, and high daily fluctuations in temperature (*Körner, Paulsen & Spehn, 2011*). The effects of

**Table 4  Pseudo-F values from pairwise PERMANOVA of relative warp scores from three anatomical aspects, between populations from different geographic subregions (sites) for jsp5 only.**

| Site | STH | WTH | CTH | SIN |
|------|-----|-----|-----|-----|
| **A. HEAD** | | | | |
| STH | 0 | | | |
| WTH | **1.743** | 0 | | |
| CTH | 8.58 | 3.622 | 0 | |
| SIN | 5.737 | 6.734 | 11.36 | 0 |
| **B. MESO** | | | | |
| STH | 0 | | | |
| WTH | 4.501 | 0 | | |
| CTH | 6.632 | 5.093 | 0 | |
| SIN | **2.461** | 5.003 | 6.495 | 0 |
| **C. PROFILE** | | | | |
| STH | 0 | | | |
| WTH | **3.161** | 0 | | |
| CTH | 5.216 | 6.789 | 0 | |
| SIN | 4.131 | 6.624 | 5.952 | 0 |

Note:
Statistically insignificant F-values—Bonferroni-corrected $p$ values ≤ 0.05—are indicated in bold. Exact $p$ values are provided in Table S7.

temperature variation, in particular, tend to be exacerbated on small-sized ectotherms such as insects (*Dahlhoff et al., 2019*).

In ants, adapting to survive in extreme habitats such as cold environments at high altitude, may involve changes to genes coding for enzymes related to key biological processes including energetic metabolism and development (*Cicconardi et al., 2020*). Such changes have been linked to the enhanced gene responses and improved abilities of cold-adapted insects to cope with hypoxia and cold stress at high elevations (*Dahlhoff et al., 2019*; *Cicconardi et al., 2020*). In this study, the genetic separation of near-sympatric populations at different elevations may therefore potentially reflect divergent adaptations to different environments. However, species-level phylogenetic signals in thermal tolerance (either heat or cold) are known to be weak or inconsistent amongst ants (*Roeder, Roeder & Bujan, 2021*). Genetic sequences used in resolving phylogenetic relationships are often not directly associated with physiological processes that influence individual organismal fitness in their natural environments. Even interpretations of variation in mitochondrial genes such as COI, which are technically linked to aerobic metabolism and hypoxia tolerance (*Dahlhoff et al., 2019*), are not straightforward. Direct links between polymorphisms in mitochondrial haplotypes and individual selection can be confounded by selection or drift acting on multiple levels—organelle, cytoplasmic, and organismal (*Rand, 2001*). Thus, divergences between all these sequences may not have developed simply due to adaptive evolution under natural selection on an individual level. Empirical data on thermal and/or hypoxic tolerances within and between known species will be

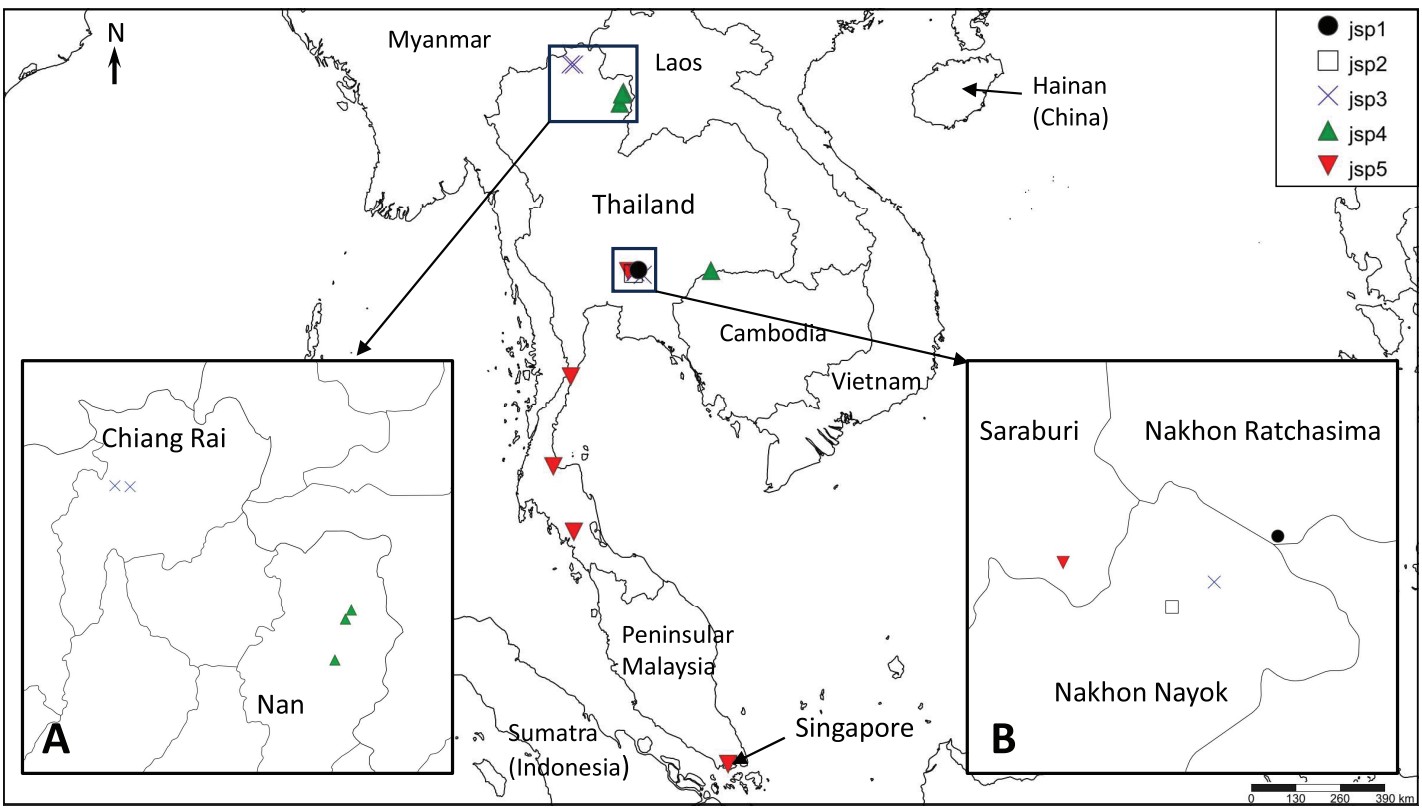

**Figure 11 Distribution map of putative _Rhopalomastix javana_ complex species jsp1-5.** (A) jsp3 and jsp4 colonies in North Thailand. (B) jsp1-3 and jsp5 colonies in Central Thailand.

pertinent in assessing the actual role of elevation (or lack thereof) in maintaining species boundaries in _Rhopalomastix_.

Elevation may also strengthen species boundaries more indirectly _via_ temperature effects on reproductive phenology and consequent asynchronous mating events of ants at different altitudes (_Roeder, Roeder & Bujan, 2021_). In general, mating seasons seem to occur later for ants at higher elevations with an average delay of one day for every 100 m increase in elevation, but this can vary amongst different species (_Helms, 2023_). Mating flights tend to be brief—a few minutes or hours (_Helms, 2023_), hence flight timings of alates from asynchronous high and low elevation colonies may never coincide. These constraints may be compounded by the minute body size and presumably poor dispersal abilities of _Rhopalomastix_ alates.

Human-mediated dispersal aside, ant dispersal distances are usually correlated with body size—alates of larger species typically fly farther than small ones (_Helms, 2018_). Moreover, flying may be even more energetically expensive for tiny _Rhopalomastix_ queens if they do engage in the strenuous carrying of symbiotic armoured scale insects (Diaspididae) from their birth colonies upon dispersal, to start diaspidid 'farms' in their new nests (see _Yong, Matile-Ferrero & Peeters (2019)_). Based on these assumptions, we posit that _Rhopalomastix_ alates are not adept at flying long distances, thereby minimizing their chances of broad range expansion or coming into contact with dispersing alates from

distant populations on their own effort. At the time of writing, hardly anything is known about the actual reproductive ecology of *Rhopalomastix*, thus these scenarios remain conjectures at best.

### Inferring intra-species population structure

Among the putative species of the *R. javana* complex, variable patterns of genetic relatedness within each species on a finer grain were apparent, based on the co-ancestry matrix estimated from MIG-seq loci. The species jsp3 and jsp5 each comprise genetically distinct allopatric populations from multiple geographic subregions, unlike the remaining species. In the co-ancestry matrix, both species exhibit diagonal blocks of higher kinship within the same subregion, albeit visibly more subtly depicted in jsp5 (Fig. 6). These diagonal patterns can be interpreted as genetic drift associated with geographic separation (*Lawson et al., 2012*), and may further imply incipient speciation. This inference is bolstered by COI clustering results: in jsp3, NTH and CTH samples split at 3.2% pairwise distance, whereas in jsp5, CTH and WTH samples diverged from STH and SIN samples at 3.5% (Fig. 4). Interspecies thresholds for COI divergence among closely-related arthropods can be as low as 2–4% (*Hebert, Ratnasingham & De Waard, 2003*; *Meier, Zhang & Ali, 2008*), though this is often not always the case.

For jsp5, STH populations sport greater admixture with SIN populations than with those from other subregions of Thailand (Fig. 6), based on co-ancestry values calculated from MIG-seq loci. This pattern is supported by ML and BI tree analyses of the same data, where CTH and WTH populations of jsp5 were recovered as well-supported clades distinct from the clade comprising SIN and STH populations (Figs. 5A, 5B). Singapore and south Thailand are geographically disjunct, separated by Peninsular Malaysia. The higher ancestry shared between these distant populations, however, may not be all that unexpected. South Thailand is mostly linked to other subregions of the country by the Isthmus of Kra—a narrow land bridge joining mainland Southeast Asia and the Thai-Malay Peninsula (*Luo et al., 2014*). The Isthmus has been widely established as a zoogeographic transition zone differentiating between terrestrial fauna of Indochina and Sundaland, though the exact transition position varies among taxa (*Hughes, Round & Woodruff, 2003*; *Woodruff & Turner, 2009*; *Luo et al., 2014*; *Hinckley et al., 2023*).

This unique zoogeographical separation is postulated to be a legacy of prehistoric sea level changes, specifically two seaways that arose across the Isthmus during the Miocene (24–23 MYA) and early Pliocene (5.5–4.5 MYA) respectively, and persisted for more than a million years each (*Hughes, Round & Woodruff, 2003*; *De Bruyn et al., 2005*). Those seaways could have been major physical barriers to dispersal and gene flow between populations from either side of the barrier for some terrestrial insects, such as ants. Over the extensive duration of separation (>1 MY), populations on opposing sides of the seaway barrier may develop exclusive intraspecific genetic signatures which could persist even if the barrier eventually fades (*De Bruyn et al., 2005*). Alternatively, the separation of fauna north and south of the Isthmus may also have resulted from more ecological factors. Wet, evergreen dipterocarp forest south of the Isthmus transitions to drier, mixed deciduous

forest in the north, at the northern end of the Isthmus (11–13°N) (*Hughes, Round & Woodruff, 2003*). Ecological differentiation in terms of host trees or habitat type between jsp5 populations north and south of the Isthmus, however, was not explicit. Most colonies of jsp5 were found nesting in lowland native trees in either evergreen or deciduous forest, regardless of their locations relative to the Isthmus. Thus, the observed genetic affinity between jsp5 populations from Singapore and STH—both considered part of Sundaland— relative to those in other Thai subregions in Indochina, is likely more a consequence of past palaeogeographic and climatic events, including sea level fluctuations, concerning the Isthmus of Kra. Further analyses with additional genetic data from specimens identifying to jsp5 from Peninsular Malaysia may be help confirm this supposition.

## Geometric morphometrics as a taxonomic tool

When used in concert with genetic methods, GM can be a powerful tool towards detecting and objectively assessing subtle variations in morphology among cryptic species, which are otherwise easily overlooked during visual examination. Based on PERMANOVA tests of relative warps in this study, we determined that every pair of putative *R. javana* complex species differed in at least one anatomical aspect—Head, Meso, Profile—with strong empirical support (Table 3). Using thin plate spline (TPS) deformation grids depicting shape changes along major principal component axes (Figs. 7–9), we could also flag taxonomically informative areas of high variation on a finer scale for potential use in species diagnosis.

Variable areas-of-interest can differ depending on the actual groups or species being assessed. In TPS analyses of Head shape, while clypeal shape appears to be highly variable between species when all or only *R. javana* species are considered (Figs. 7B, 7C), the latter group also seems to vary intensely in terms of eye size and the median concavity of the posterior head margin (see darker-coloured areas in Fig. 7C). Furthermore, similar analyses of Meso shape reveal areas of high variation exclusive to the *R. javana* complex: curvature of dorsolateral margins of propodeum (Fig. 8C) and general shape of promesonotum (Fig. 8D).

Besides quantifying morphological differences between cryptic species, we can estimate the diagnostic reliability of each target shape by calculating classification accuracy based on canonical variates. These results enable taxonomists to identify and prioritise taxonomically more informative characters for deeper scrutiny. In this study, we found classification accuracy to be highest for Head shape (all species—80.8%; *R. javana* species—77.8%) and lowest for Meso shape (all species—60.7%; *R. javana* species—60.6%), regardless of whether all or only *R. javana* species were considered. This suggests that amongst all three anatomical aspects, Head shape may be the most robust character for discriminating between species of *Rhopalomastix*.

Head structure and its underlying musculature are closely connected to the bark-nesting ant's ability to chew tunnels through wood. In addition, the head of *Rhopalomastix* also houses silk-producing glands and controls processes of silk secretion (*Yong, Matile-Ferrero & Peeters, 2019*; *Billen & Peeters, 2020*). Because of its direct association with physiological processes that affect evolutionary fitness in *Rhopalomastix*, the head as a whole may be

subject to stronger selective pressures and consequently higher evolutionary rates relative to other body parts. This may partly explain why Head shape was found to be most informative and reliable in species discrimination for the genus. Additional experimental evidence and/or fine-scale habitat information, are required to verify these speculations.

## Limitations of GM

Taxonomic utility notwithstanding, significant differences in shape or structure detected through GM analyses alone do not necessarily constitute incontrovertible proof of species separation. Procrustes ANOVA of superimposed landmark coordinates in this study showed significant effects of geographic location—represented by the factor 'site' in the model—on shape variation, though notably weaker than species effects. Allopatric populations of the same species expressing sufficient morphological variation associated with genetic drift, may also be verified as distinct entities by GM approaches. We see this happening in pairwise PERMANOVA tests comparing shapes from different geographic sub-groups within jsp3 and jsp5 (Table 4) respectively. Although not every pair of geographic sub-groups was statistically dissimilar, the mixed results reinforce the notion that other factors such as geography may confound direct species effects on empirical differences in morphology.

The same logic applies otherwise—the absence of differences among groups by itself is not an unequivocal sign of conspecificity. The effectiveness of GM as a taxonomic tool is largely contingent on the target shape/s selected for quantification. Negligible differences between analysed shapes simply leave open the possibility that differences may persist in other external traits—not only other bodily structures, but also features such as sculpture and pilosity which are not quantifiable by GM methods. Moreover, it is also possible that genetic heterogeneity among species may not always translate to quantifiable morphological variability, particularly when 'neutral' traits not explicitly subject to natural selection are involved.

The limitations of GM are perhaps most strikingly illustrated in comparisons of shape between two divergent species pairs: (1) jsp2 and *R. glabricephala*, and (2) the outgroup species—*R. glabricephala* and *R. murphyi*. Each pair comprises indisputably distinct species delimited by large COI divergence thresholds—19.2% uncorrected pairwise distance respectively (Fig. 4). However, both species in each pair turned out statistically undifferentiated in all three anatomical aspects (Table 3). The species can instead be distinguished by differences in non-shape characters that are impossible to detect using GM. For instance, head dorsum in jsp2 is striated as opposed to smooth and shiny in *R. glabricephala* (*Wang, Yong & Jaitrong, 2018*). For the two outgroup species, the anterior face of petiole has decumbent hairs in *R. glabricephala*, but hairs are absent in *R. murphyi* (*Wang, Yong & Jaitrong, 2018*).

Pronounced intra-species variations in body shape and/or structure are also essential points to consider in planning GM analyses. For some ants, strong allometric effects on body shape can occur with increasing body size, especially in workers, confounding

assessments and comparisons of mean shape for the affected body parts. Genera with morphologically distinct major and minor castes, such as *Pheidole* and *Carebara*, best demonstrate such phenomena. In contrast, little to no allometric effects of size on the three standard anatomical aspects, *i.e.*, Head, Meso, Profile, have been observed in monomorphic workers of the *R. javana* complex (*Wang, Yong & Jaitrong, 2021*). Nevertheless, for species sporting explicit allometric variation, we do recommend conducting separate sets of GM analyses for different size ranks. Alternatively, a larger number of specimens capturing broad allometric variation may be sampled, but it is often difficult to achieve equal representation across size classes.

Given the inherent caveats of GM (or any other morphology-based method), its advantages are probably best exploited when applied to complement or test species hypotheses initially built on genetic evidence. The utility of GM in assessing species boundaries may be augmented by selecting characters more explicitly linked to reproductive isolation—a definitive criterion for species status—for analysis. For example, male genitalia can be analysed with GM, as has been done for the giant ant *Dinoponera* (see *Tozetto & Lattke (2020)*). However, analyses of such sexually-selected characters also have to be interpreted with caution. Recent studies have demonstrated significant intraspecific variation in male genitalia of other insects, consequences of phenotypic plasticity under environmental influences such as altitude (*Kiss, Toth & Varga, 2017*) and seasons (*Fumi & Friberg, 2019*).

## CONCLUSIONS

Despite the drawbacks, it is undeniable that combining multiple lines of evidence from different methods of taxonomic analysis allows us to gain broader insights on evolutionary relationships and build more robust species hypotheses, especially where cryptic species are concerned. Our exploration of the inscrutable *R. javana* complex demonstrates how a multi-pronged approach might be a, if not the only, panacea towards reducing ambiguities in resolving cryptic diversity for ants. Specifically, our success with a combined DNA-GM approach—albeit limited to a relatively small sample size—corroborates the use of geometric morphometrics in complement to genetic data to facilitate taxonomic resolution of cryptic ant species. Analyses of genetic data enabled us to infer plausible patterns of population structure and probable present species statuses. Quantitative analyses of morphology *via* geometric morphometrics helped us overcome the limits of visual judgement, and detected subtle but sound differences within and between putative species. Informed by these results, we are better placed to refine species diagnoses in future formal descriptions of the five cryptic lineages recovered in this study.

The evident utility of a combined DNA-GM workflow aside, many gaps of knowledge in our understanding of the biology and ecology of *Rhopalomastix* prevent clarification of species boundaries in the *R. javana* complex. For example, more precise definitions of the ecological niches occupied by each *R. javana* population are required to support hypotheses of genetic species boundaries maintained by niche divergence. In particular, we

need to understand the reproductive phenology of *Rhopalomastix*, and how it may vary in different environments. Ideally, we should also investigate host tree preferences and their potential implications on species distributions. We could not properly evaluate the host tree aspect in this study because most host trees were unidentified. In addition, targeted sequencing of suites of the genome encoding specific metabolic enzymes in each putative species may help identify genetic signatures explicitly linked to divergent environmental adaptations (*Cicconardi et al., 2020*).

In follow-up to this research, we plan to conduct a more comprehensive taxonomic investigation into the *R. javana* complex, establishing the five genetic lineages recovered here—jsp1-5—as formal named species. For this purpose, we aim to obtain more colony samples ideally covering a broader geographic range. Genetic species identities of these new samples will be determined *via* COI barcode clustering or matching. Subsequently, we will conduct definitive characterisation of inter-species morphological differences by close examination of representative specimens from all samples. Besides the standard morphological measurements unique to ant taxonomy, we shall pay particular attention to the bodily areas-of-interest with relatively high inter-specific variation, *e.g.*, clypeal shape, that were initially highlighted in this study. Both intra- and inter-species variation in such external traits such as clypeal shape shall be quantified or qualified in precise terms that enable practical species diagnoses. Regardless of the many issues that remain unaddressed and unanswered questions, this study represents a small but indispensable first step in the larger blueprint towards demystifying cryptic diversity in *Rhopalomastix* and perhaps, other similarly inscrutable ants.

## ACKNOWLEDGEMENTS

We would like to thank James Rohlf for his kind and swift help with tps software download issues, also for making the excellent tps series freely available online. We also thank Danwei Huang, Katsuyuki Eguchi and Emiko Oguri for allowing use of lab resources and facilities to perform MinION barcoding and MIG-seq procedures respectively. We thank Marc Chang for assistance with library preparation and downstream bioinformatic processing of MinION reads. We are grateful to Weeyawat Jaitrong for facilitating the acquisition and use of colony samples from Thailand, and Gordon Yong for help with field collection of *Rhopalomastix*. Finally, we deeply appreciate the valuable and constructive comments made by the unnamed reviewers that helped us improve on earlier versions of the manuscript.

### Funding

The authors received no funding for this work.

### Competing Interests

The authors declare that they have no competing interests.
## Author Contributions

- Wendy Y. Wang conceived and designed the experiments, performed the experiments, analyzed the data, prepared figures and/or tables, authored or reviewed drafts of the article, and approved the final draft.
- Aiki Yamada performed the experiments, analyzed the data, authored or reviewed drafts of the article, and approved the final draft.

## DNA Deposition

The following information was supplied regarding the deposition of DNA sequences:

The COI barcode sequences are available in the Supplemental File and at GenBank: OR262214–OR262321.

The demultiplexed raw MIG-seq reads are available at Zenodo: Wang, W., & Yamada, A. (2023). MIG-seq data of *Rhopalomastix javana* complex (1.0) [Data set]. Zenodo. https://doi.org/10.5281/zenodo.8177513.

## Data Availability

The COI barcode sequences are available in the Supplemental File and at GenBank: OR262214–OR262321.

The demultiplexed raw MIG-seq reads are available at Zenodo: Wang, W., & Yamada, A. (2023). MIG-seq data of *Rhopalomastix javana* complex (1.0) [Data set]. Zenodo. https://doi.org/10.5281/zenodo.8177513.

The digitized landmarks and semilandmarks for each of three anatomical aspects are available in the Supplemental File.

## Supplemental Information

Supplemental information for this article can be found online at http://dx.doi.org/10.7717/peerj.16416#supplemental-information.

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
