# Peer review of "Scrutinising an inscrutable bark-nesting ant: Exploring cryptic diversity in the Rhopalomastix javana (Hymenoptera: Formicidae) complex using DNA barcodes, genome-wide MIG-seq and geometric morphometrics"

_PeerJ, doi:10.7717/peerj.16416_

## Round 0.1 · original submission · Minor Revisions

The reviewers provide extensive thoughtful comments which you should address. One of the more significant comments voiced by all three reviewers is that the conclusions are not clear with respect to the demarcation of species. Reviewer 3 suggests that perhaps this is intentional and the authors intend to prepare a follow-up paper describing the new species. If this is the case, that is perfectly acceptable but please state your intentions in this paper so it is clear to readers. Regardless, it must be made clear in this study which are the populations likely deserving of species status. A reviewer also raises concerns about the lack of images of the chosen landmarks which could affect the repeatability of the study. Please address all of these concerns in your revision.

Reviewer 1 ·

Basic reporting

-Interesting approach to integrative taxonomy. Good introduction about incorporating geometric morphometrics into ant taxonomy. I would add more information about why it is important to investigate the species boundaries in R. javana. Why is this organism of interest to biology?

-It is not clear if the study meets the main objectives. The main point of the paper is to investigate the species boundaries in R. javana, but at the same time, the paper does not state a clear conclusion on how many and which are the species in R. javana. I think this makes the take-home message of the study confusing to the reader.

-Detailed description of the methods developed in the study.

-Discussion: You suggest that elevation might cause divergence between species. However, did you conduct species delimitation analyses using elevation as a discriminatory factor? what would be the speciation mechanism in such a case? would you treat them as distinct or the same species? do the groups delimited by morphometrics or phylogenetics correspond to differences in elevation?

139: These include three colonies that were morphologically identified as: R. glabricephala, R. murphyi, and tentatively R. murphyi, respectively.

190: double parenthesis.

Figures 1,2,3 can be combined into a single figure.

Figure 4: The ML inference indicates low support for the split of several lineages. In contrast, the Bayesian inference presents much better-supported values. How would you reconcile these two? Which one is guiding your species delimitation?

-How deep is the divergence between those inferred clades? I think the study would benefit from a divergence dating analysis.

-Interesting co-ancestry matrix! How would you interpret the values in the matrix? is there hybridization happening between some of the groups?

-Please check the format of the references.

Experimental design

Comprehensive use of geometric morphometrics. Interesting research question. Good description of the methods used.

For improvement:
- Include a divergence dating analysis
- Test for isolation by elevation

Validity of the findings

The results are very interesting and the authors include a good discussion on the importance of using geometric morphometrics in taxonomic studies. This is a well-conducted study with comprehensive analyses included. However, the implications of the results need to be explored deeper. With the evidence we have so far, what can we conclude? The main conclusion of the study is not clearly stated. It is not clear which and how many species were delimited in the study. The hypotheses and conclusions need to be revised.

Additional comments

Very interesting study and important incorporation of geometric morphometrics into ant taxonomy!

Reviewer 2 ·

Basic reporting

This article tackles the cryptic diversity of Rhopalomastix ants focusing on ants currently assigned to the species R. javana and with a geographical focus on Thailand. Cryptic diversity is relevant for assessing biodiversity as of course, animals do not need to play by the long-standing rules of taxonomy. And further, clearly identifying and classifying patterns of cryptic diversity through multiple lines of evidence and an integrative workflow allows more insight into the evolutionary process.
Here, the authors elegantly follow an integrative workflow applied to collection material to identify distinct Rhopalomastix lineages within the current R. javana and they find evidence to explain the evolutionary scenarios leading to the structure they observe.

Basically, this article already represents a valuable contribution. I would like to make some suggestions below in this field, and ask the authors to clarify concerns I have upon reading regarding their data analysis design and their interpretations in the other boxes.
1. Motivation for conducting this analysis
I think within the Introduction, first and last paragraph, the authors could elaborate more on why they conducted this analysis, ideally beyond the ant genus Rhopalomastix. Understanding patterns in cryptic diversity necessitates rethinking species concepts as indicated by the authors' Fiser et al. 2018 citation. However, I'd argue it also has severe implications worth mentioning beyond that, e.g. in the context of the Anthropocene and biodiversity loss. Moreover, the presence of cryptic diversity in ants itself could be highlighted more. Although this journal does not require the authors to fish for higher impact, I think it cannot hurt to put the research in a wider context. Future readers could highly benefit from that.
2. Visualizing the biogeographic structure
Following the article, biogeography likely plays an essential role towards understanding R javana. I think a map showing where the samples were collected, maybe even the topographic information of high- vs lowlands would really benefit the paper.
3. Implications of jsp1 and jsp2
What does it mean that the authors recovered single colonies that may represent distinct species within their sample? The discussion lacks an elaboration here. Can we expect more cryptic diversity if we were to sample central Thailand more? Do these represent remnants of an ancestral population whereas jsp3-5 went through a radiation and range expansion? Can we draw anything from that pattern at all?
4. Unnecessary "Limitations of GM" subchapter
The authors come to the conclusion that GM can be a complementary tool for taxonomy in general, but cannot stand on its own within the field. I think that 4 paragraphs on that topic is excessive and that this whole section could be condensed and well incorporated into the previous subsection.

According to the journal's guidelines, I would further like to state that for the GM analysis, there was no indication of images or landmark files provided, which makes potential reproduction of the results more difficult.

Experimental design

The authors follow a rigorous workflow and lay out in great detail the methods they implemented to conduct their study. I have three remarks towards the GM workflow design, where I would like the authors to clarify.

1. What is the effect of duplicating specimens?
From the results of the GM analyses, it seems that the authors left all duplicated images in the final dataset and that the statistical values also include those. Would omitting or taking shape averages of individual ant specimens affect the results in any way?

2. Why this sampling and density of semilandmarks
Why did the authors choose these 13 landmarks, is it the maximum possible? Why did the authors choose to sample the semilandmarks with that exact number of points? Are the overlapping landmarks analysed as separate landmarks when in fact, they represent the same point?

3. Did the authors consider correcting for symmetry?
As Head and Meso are bilaterally symmetrical, did the authors consider enforcing bilateral symmetry here, did they test for not doing so on the results? Did the authors consider only analysing half of the landmark set?

Validity of the findings

All results are either robust or carefully discussed. However, given the duplication of sample size for the GM analysis, I would question those results and inferences drawn from that.

Reviewer 3 ·

Basic reporting

This paper is written in clear and professional English. The introduction covers the necessary background information and clearly states the aim of the paper. The figures and data are available, relevant and of high quality.
>It would be helpful to add a figure with a distribution map for the different putative species, especially for readers not familiar with the geography of the study area. It may even replace Table 1, which could be moved to the supplementary material.

Experimental design

The paper is focused on a single question, whether or not cryptic diversity is present in R. javana. This question is novel and relevant to ant taxonomists and the methods are well described and appear to fully allow replication.The research question is addressed using two types of DNA data (mtDNA and MIG-seq data) and geometric morphometrics (GM). This approach is adequate and investigations are performed according to high standards. However the sample size of 25 colonies is rather low for assessing the assessment of 5 putative cryptic species, 2 of which were represented by only a single colony. Due to the rather careful interpretations of the authors, this is OK.

Validity of the findings

The main finding, the presence of cryptic diversity in R. javana, is supported very convincingly by agreement between twoDNAdata sources of high quality. The relevance of GM is less clear, but it is still informativeand well done. The work is transparent and all data can beaccessed by readers. The conclusions are justified and the authors are even rather careful with their interpretations. For example, they conclude that R. javana comprises “more than one species” (l. 635-637), without stating the exact number of species.
>GM part: body shape may be biased by allometric effects, especially when sample sizes are low. This bias should either be corrected for or discussed as a potential limitation.

Additional comments

This is an interesting and valuable paper about cryptic diversity in Rhopalomastix ants! I definitely recommend it for publication, with only few minor adjustments. There are some obvious questions which are not addressed at all (e.g., general morphology, diagnosis, taxonomic history, ecology), but this is OK, as otherwise the paper may become overloaded. However, I strongly encourage the authors to write a follow-up publication in the form of a taxonomic revision, which should ideally be done with a larger sample size.
>In the introduction, GM is presented as superior to conventional morphometrics (l. 68-70, 81-85). However, the insights from GM reported in this studyseem rather disappointing when compared to most studies using conventional morphometrics. There is high classification error in all aspects. It is more difficult to convey an overall shape than single measurements and we do not learn what the body shapes look like for the 5 putative species. Thus, the study does not provide a concrete solution for taxonomists as to how these lineages can be separated by shape. These limitations should be addressed or at leastdiscussed.
>l. 93-94: “no study (at the time of writing) has yet explored the combined DNA-GM approach in ant taxonomy” - this has been done before (e.g., Wagner et al. 2017, Myrmecological News 25, 95-129).
>Table 1: consider adding geographic coordinates.
>legend of Table 3: “Statistically insignificant F-values” - do you mean significant values?
>Figure 4 & 5: consider to remove the yellowish background color for the outgroup taxa. This creates the wrong impression that these are also a group/species. It is also initially confusing because we expect to see 5 cryptic species and then there are 6 groups visualized in the trees.

---

## Round 0.2 · accepted · Accept

The authors have addressed all of the reviewers' comments and the manuscript is now ready for publication. This is an excellent study and I look forward to seeing the subsequent taxonomic paper published.